# Effect of any form of steroids in comparison with that of other medications on the duration of olfactory dysfunction in patients with COVID-19: A systematic review of randomized trials and quasi-experimental studies

Phyo Zin Winn[1]*, Thein Hlaing[2], Kyaw Myo Tun[3], Seim Lei Lei[4]

1 Community-initiated COVID-19 Response Project, Myanmar Health Assistant Association, Magway Region, Myanmar, 2 District Public Health Department (Ministry of Health), Pyay District, Pyay, Bago Region, Myanmar, 3 Department of Health and Social Sciences, STI Myanmar University, Yangon, Myanmar, 4 Community Initiative COVID-19 Response Project, Myanmar Health Assistant Association, Sagaing Region, Myanmar

* mg.phyozinwinn@gmail.com

**Data Availability Statement:** All relevant data is within the paper and its Supporting Information files.

## Abstract

### Background

The life quality of about two-thirds of patients with COVID-19 is affected by related olfactory dysfunctions. The negative impact of olfactory dysfunction ranged from the decreased pleasure of eating to impaired quality of life. This research aimed to provide a comprehensive understanding of the effects of corticosteroid treatments by comparing that to other currently available treatments and interventions.

### Methods

The Preferred Reporting Items for Systematic Reviews and Meta-Analyses (PRISMA) checklist's 27-point checklist was used to conduct this review. PubMed (Public/Publisher MEDLINE), PubMed Central and EMBASE (Excerpta Medica Database) databases were conveniently selected and Boolean search commands were used for a comprehensive literature search. Five core search terms were "effects of treatments", " COVID-19-related olfactory dysfunction", "corticosteroids", "treatments" and "interventions". The reporting qualities of the included studies were appraised using JBI (Joanna Briggs Institute) appraisal tools. The characteristics of the 21 experimental studies with a total sample (of 130,550) were aggregated using frequencies and percentages and presented descriptively. The main interventions and their effects on the duration of the COVID-19-related olfactory dysfunction were narratively analyzed.

**Funding:** The authors received no specific funding for this work.

**Competing interests:** The authors have declared that no competing interests exist

## Results

Among patients with COVID-19, the normal functions of the olfactory lobe were about 23 days earlier to gain with the treatments of fluticasone and triamcinolone acetonide nasal spray compared with that of mometasone furoate nasal spray and oral corticosteroid. The smell loss duration was reduced by fluticasone and triamcinolone acetonide nasal spray 9 days earlier than the inflawell syrup and 16 days earlier than the lavender syrup. The nasal spray of corticosteroids ended the COVID-19-related smell loss symptoms 2 days earlier than the zinc supplementation, about 47 days earlier than carbamazepine treatment and was more effective than palmitoylethanolamide (PEA) and luteolin and omega-3 supplementations and olfactory training. Treatment with oral corticosteroid plus olfactory training significantly improved Threshold, Discrimination and Identification (TDI) scores compared with olfactory training alone. A full dose of the COVID-19 vaccination was not uncertain to reduce the COVID-19-related smell loss duration.

## Conclusion

Corticosteroid treatment is effective in reducing the duration of COVID-19-related smell loss and olfactory training, the basic, essential and effective intervention, should be used as a combination therapy.

## Introduction

World Health Organization (WHO) declared the coronavirus outbreak a global health emergency on 30 January 2020. In the year 2022, the pandemic still exists, and the coronavirus mutates frequently from one variant to another prolonging the length of a global health emergency. The current coronavirus (COVID-19) can cause severe pneumonia in susceptible persons, and lead to the death of patients. Therefore, it is called severe acute respiratory syndrome coronavirus 2 (SARS-CoV-2) which causes multi-organ inflammation in its most severe form [1–3]. The SARS-CoV-2 virus enters the respiratory tract of humans through the angiotensin-converting enzyme 2 (ACE-2) receptors playing a crucial role in multiorgan dysfunction. Among manifestations of multiple organ dysfunction syndromes (MODS) caused by COVID-19, the loss of taste or smell is very common and the earliest warning sign resulting from sensory system dysfunction. During the COVID-19 pandemic, many people infected with the COVID-19 virus became defected their olfactory functions and lost their important quality of life [4].

Currently, anosmia, the loss of smell mainly caused by COVID-19 infection can be partial or complete and temporary or permanent [4], but it may be due to other several causes like allergies, common cold, and serious brain or nerve damage [5]. Some similar terms have close meanings to anosmia like parosmia and hyposmia. Parosmia is the disorder of smell function in which the taste of smell is distorted [6]. Hyposmia is the reduction of the sense of smell which may be due to nasal congestion or inflammation of the nasal lining [7]. Olfactory function is, in fact, a protective mechanism for the human body from environmental hazards and microorganisms [8]. From the past two years till now, the consequences of COVID-19-induced olfactory dysfunction highly impacted many lives. The reported prevalence of loss of smell among patients with COVID-19 in various countries was 98.0% in Iran, 91.3% in Ukraine, 73.6% in Germany, 72.0% in Poland, 67.9% in Egypt, 61.7% in Turkey, 58.1% in the UK (United Kingdom), 57.2% in Dominican, 56.9% in Chile, 54.7% in Guatemala, 50.8% in

Brazil, 47.2% in Mexico, 47.0% in France, 46.7% in the US (United States), 41.7% in Italy, 41.3% in Pakistan, 40.4% in Ecuador, 37.6% in Honduras, 32.7% in Bolivia, 32.0% in China, 31.9% in India, 23.9% in Peru, 15.3% in South Korea and 12.0% in Canada [9–22].

Some proposed mechanisms of anosmia include olfactory cleft obstruction, local inflammation in the nasal epithelium, early apoptosis of olfactory cells, changes in olfactory cilia, injury to the olfactory epithelium, and damage to the olfactory neurons or stem cell neurons [23]. The health hazards from loss of olfactory function are that the patient may get gastroenteritis from ingestion of spoiled foods, a decrease in the pleasure of eating and drinking, changes in eating habits, and impairments of psychological conditions which in turn are associated with immune dysfunction. Moreover, people with olfactory dysfunction are unable to detect their own body smell, making them weak in personal hygiene and social communication. The olfactory disorder is the potential to be risky for occupational health hazards and other safety issues because of their inability to detect harmful odours and to be low self-confidence which can lead to depression and greater reductions in quality of life [24–27].

Several interventions exist for the treatment of olfactory disorders, which differ depending on aetiology in terms of medical, surgical, physical and counselling interventions [28–32]. Since the mechanism of anosmia is not known yet, the treatment options are also varied according to the perception of the physician. Some give neuro-regenerative drugs like cerebrolysin and vitamin B complex while others administer steroids in systemic form or inhaled form [33]. The current treatments given for this olfactory dysfunction by various medical personnel include systemic steroid tablets, internasal steroids, cerebrolycin, gabapentin, vitamin B complex, intranasal vitamin A, etc. [34]. However, it has not been known which treatment option exerts the best response to olfactory dysfunction in patients with COVID-19. This review aimed to summarize all possible treatment options and other interventions relieving COVID-19-related anosmia and determine the effectiveness of steroids on the treatment and the duration of the COVID-19-related olfactory dysfunction compared to that of other medications and interventions.

## Material and method

### Study design

According to the PICOT criteria (Table 1), the predefined research question was: "How the corticosteroid treatment options are more effective in reducing the duration of COVID-19-related olfactory dysfunction as compared with other types of interventions?". This review

**Table 1. The frame of review question.**

| Model Frame | Criteria | Definitions |
|---|---|---|
| P | Patients | Those who were infected with severe acute respiratory syndrome coronavirus 2 (SARS-CoV-2) including new variants such as B.1.1.529, Alpha (B.1.1.7), Beta (B.1.351), Gamma (P.1), Delta variant, Omicron variant, Omicron "stealth" variant (BA.1, BA.2). |
| I | Intervention | Steroid treatments, other treatments and interventions for the COVID-19-related olfactory dysfunction |
| C | Comparison | Between treatment/intervention groups and placebo groups and between steroid treatments and other treatments/interventions as well as before and after treatments/interventions for the COVID-19-related olfactory dysfunction |
| O | Outcome | The shorter or longer duration between onset and recovery from the COVID-19-related odour loss |
| T | Time | During the COVID-19 pandemic (After 2019) |

was conducted according to the 27-point checklist of the Preferred Reporting Items for Systematic Reviews and Meta-Analyses (PRISMA) checklist [35].

## Typical inclusion criteria

**Study objective, publication date and study type.**   This review was more likely to analyze the outcomes resulting from treatment/intervention trials that emphasized the effects of treatment options for COVID-19-related olfactory dysfunction and therefore included the relevant trial works published during the COVID-19 pandemic. This review did not consider any trial for the treatment of smell loss due to other causes (e.g., Nasal polyps, Common cold, Influenza, Sinusitis, other sino-nasal abnormalities and surgery-associated causes) rather than the COVID-19 infection. The study designs of the included studies were to be clinical trials such as randomized controlled trials and quasi-experimental studies and test a specific drug or other treatment options on several same eligible people.

**Publication status.**   The reviewers considered the primary studies printed in English only to be included. The published papers were to be full-texts, available online, downloadable, and free access. Importantly, this review excluded such publication types as abbreviations, abstract forms, dissertation reports, book publications, advertisements, anecdotes, review reports, case reports, clinical conference reports, editorial reports, guidelines, literature reviews, systematic reviews, systematic reviews and meta-analyses, letters, meeting abstract reports, news, clinical instructions, clinical outlines, and patient education handouts.

**Study population.**   The study participants had to

a. Be infected with the COVID-19 virus

b. Be mild or moderate and polymerase chain reaction (PCR)-confirmed SARS-CoV-2infection (Mild Covid-19 was defined as an individual who has one or more COVID-19-related signs and symptoms such as fever, nausea, vomiting, malaise, diarrhoea, taste and smell loss and cough without breathlessness and others abnormal chest imaging. Moderate COVID-19 was defined as an individual with the manifestations of mild COVID-19 plus abnormal chest imaging, lower respiratory disease, and pulse oximeter (saturation of peripheral oxygen [SpO2] $\geq$94%). The definition of a severe form of COVID-19 included $SpO_2$ <94%, a respiration rate per minute more than 30, a percent of lung infiltrates more than 50 and arterial partial pressure of oxygen (PaO2)/ Fraction of Inspired Oxygen (FiO2) less than 300 mmHg.)

c. Be confirmed with a symptom of the COVID-19-related loss of smell

d. Be adults

The study participants had not to

a. Be pregnant

b. Be severe patients with COVID-19

c. Suffer from psychological disorders

d. Develop loss of smell before the COVID-19 infection

e. Be associated with a history of sino-nasal abnormalities, diseases or surgeries

**Intervention.**   In this review, interventions were any form of treatment options used for the COVID-19-related loss of smell and the researcher expected such treatment interventions

as oral, intranasal and systemic corticosteroids, intranasal sodium citrates, smell training, essential oil, diets, supplements, intranasal Vitamin A, systemic omega-3, coffee, lavender syrup, vaccination, inflawell syrup, carbamazepine and palmitoylethanolamide (PEA) plus luteolin.

**Control group.**   If the included studies recruited the control groups, their allocations had to be unbiased according to the sample size determinations considering the power and significance level. Otherwise, this review picked the quasi-experimental studies that assigned the control groups without randomization.

**Outcomes.**   The outcomes of the included studies had to investigate the effects of corticosteroid medication, other medications and the specific interventions on the duration of COVID-19-related olfactory dysfunction. The outcomes had to include the time to clinical improvements from the COVID-19-related olfactory dysfunction especially loss of smell following treatment interventions.

**Information sources and search strategy.**   The reviewers searched three medical databases (PubMed, PubMed Central and EMBASE) from 21 March 2022 to 30 April 2022 for the eligible studies relevant to the predefined inclusion criteria. In this review, "Effects of treatments", "COVID-19-related olfactory dysfunction", "Corticosteroids", "Treatments" and "Interventions" were five core search terms. The phrases, keywords and other similar terms necessary for a precise search were collected from much literature and previous studies and for which the PubMed Advanced Search Builder was also applied.

In searching the eligible studies in PubMed and PubMed Central, the following search terms were applied by two reviewers.

*"Treatments" or Interventions" and "COVID-19-related olfactory dysfunction"*. "Treatment"[Mesh] OR "Treatment"[All Fields] OR "Medication"[All Fields] OR "Therapy"[All Fields] OR "Administration"[Mesh] OR "Management"[All Fields] OR "Care of"[All Fields] OR "Cure"[Mesh] OR "Prescription"[All Fields] OR "Regimen"[All Fields] OR "Oral corticosteroid"[All Fields] OR "Intranasal corticosteroid"[All Fields] OR "Systemic corticosteroid"[All Fields] OR "Intranasal sodium citrate"[All Fields] OR "Smell training"[All Fields] OR "Taste training"[All Fields] OR "Olfactory training"[All Fields] OR "Essential oil"[All Fields] OR "Diets"[All Fields] OR "Supplements"[All Fields] OR "Intranasal Vitamin A"[All Fields] OR "Systemic Omega-3"[All Fields] OR "Coffee"{All Fields] OR "Lavender Syrup"[Mesh] OR "Lavender Syrups"[All Fields] OR "Vaccination"[Mesh] OR "Inflawell Syrup"[Mesh] OR "Inflawell Syrup"[All Fields] OR "Carbamazepine"[Mesh] OR "Palmitoylethanolamide"[Mesh] OR "PEA"[Mesh] OR "Luteolin"[Mesh] AND " COVID-19-related olfactory dysfunction"[Mesh] OR " COVID-19-related olfactory dysfunctions"[TIAB] OR " COVID-19-associated olfactory dysfunction"[TIAB] OR "COVID-19-related smell loss"[Mesh] OR "COVID-19-related odor loss"[All Fields] OR "COVID-19-related loss of smell"[All Fields] OR " COVID-19-related loss of odor"[All Fields] OR COVID-19-associated smell loss"[Mesh] OR "COVID-19-associated odor loss"[All Fields] OR " COVID-19-associated loss of smell"[All Fields] OR " COVID-19-associated loss of odor"[All Fields] OR "anosmia"[All Fields] OR "parosmia"[All Fields] OR "olfactory dysfunction"[All Fields] OR "olfactory disorder"[All Fields]

*"Effects of Treatment" and "Corticosteroid" and "COVID-19-related olfactory dysfunction"*. "Effects of Treatment"[Mesh] OR "Effectiveness of Treatment"[All Fields] OR "Treatment effectiveness"[All Fields] OR "Treatment efficacy"[All Fields] OR "Strengths of Treatment"[All Fields] AND "Corticosteroid"[Mesh] OR "Steroid"[All Fields] OR "Steroids"[All Fields] OR "Delivering Corticosteroid treatment"[Mesh] OR "Administrating Corticosteroid treatment "[All Fields] OR "Indexing Corticosteroid treatment "[All Fields] OR "Dosing Corticosteroid treatment "[All Fields] OR "Delivering Steroid treatment"[All Fields] OR "Administrating Steroid treatment "[Mesh] OR "Indexing Steroid treatment "[All Fields] OR "Dosing Steroid

treatment "[Mesh] AND "COVID-19-related olfactory dysfunction"[Mesh] OR "COVID-19-related olfactory dysfunctions"[TIAB] OR "COVID-19-associated olfactory dysfunction"[TIAB] OR "COVID-19-related smell loss"[Mesh] OR " COVID-19-related odor loss"[All Fields] OR "COVID-19-related loss of smell"[All Fields] OR "COVID-19-related loss of odor"[All Fields] OR "COVID-19-associated smell loss"[Mesh] OR " COVID-19-associated odor loss"[All Fields] OR "COVID-19-associated loss of smell"[All Fields] OR "COVID-19-associated loss of odor"[All Fields] OR "anosmia"[Mesh] OR "parosmia"[Mesh] OR "olfactory dysfunction"[Mesh] OR "olfactory disorder"[Mesh]

In searching the includable studies in EMBASE, the Quick Search and PICO Search were exercised. For a quick search, the following key terms were applied.

'Treatments' or 'Interventions' and 'COVID-19-related olfactory dysfunction'. 'Treatment'/de OR 'treatment' OR 'tx' OR 'Medication' OR 'Therapy' OR 'Administration' OR 'Management' OR 'Mx' OR 'Care of'/exp OR 'Cure' OR 'Prescription'/exp OR 'Regimen' OR 'Oral corticosteroid'/exp OR 'Intranasal corticosteroid' OR 'Systemic corticosteroid' OR 'Intranasal sodium citrate'/exp OR 'Smell training'/exp OR 'Taste training' OR 'Olfactory training' OR 'Essential oil'/exp OR 'Diets'/exp OR 'Supplements'/exp OR 'Intranasal Vitamin A'/exp OR 'Systemic Omega-3' OR 'Coffee' OR 'Lavender Syrup' OR 'Vaccination' OR 'Inflawell Syrup' OR 'Carbamazepine' OR 'Palmitoylethanolamide'/PEA OR 'Luteolin' AND 'COVID-19-related olfactory dysfunction'/de OR 'COVID-19-related olfactory dysfunctions'/exp OR 'COVID-19-associated olfactory dysfunction'/exp OR 'COVID-19-related smell loss'/exp OR 'COVID-19-related odor loss'/exp OR 'COVID-19-related loss of smell'/exp OR 'COVID-19-related loss of odor'/exp OR 'COVID-19-associated smell loss'/de OR 'COVID-19-associated odor loss'/exp OR 'COVID-19-associated loss of smell'/exp OR 'COVID-19-associated loss of odor'/exp OR 'anosmia'/de OR 'parosmia'/exp OR 'olfactory dysfunction'/de OR 'olfactory disorder'

"Effects of Treatment" and "Corticosteroid" and "COVID-19-related olfactory dysfunction". 'Effects of Treatment'/de OR 'Effectiveness of Treatment' OR 'tx eff' OR 'Treatment effectiveness' OR 'Treatment efficacy' OR 'Strengths of Treatment' AND 'Corticosteroid'/de OR 'Steroid'/exp OR 'Steroids'/exp OR 'Delivering Corticosteroid treatment'/de OR 'Administrating Corticosteroid treatment'/exp OR 'Indexing Corticosteroid treatment'/exp OR 'Dosing Corticosteroid treatment'/exp OR 'Delivering Steroid treatment'/exp OR 'Administrating Steroid treatment'/exp OR 'Indexing Steroid treatment'/exp OR 'Dosing Steroid treatment'/exp AND 'COVID-19-related olfactory dysfunction'/de OR 'COVID-19-related olfactory dysfunctions'/exp OR 'COVID-19-associated olfactory dysfunction'/exp OR 'COVID-19-related smell loss'/exp OR ' COVID-19-related odor loss'/exp OR 'COVID-19-related loss of smell'/exp OR 'COVID-19-related loss of odor'/exp OR 'COVID-19-associated smell loss'/de OR 'COVID-19-associated odor loss'/exp OR 'COVID-19-associated loss of smell'/exp OR 'COVID-19-associated loss of odor'/exp OR 'anosmia'/de OR 'parosmia'/exp OR 'olfactory dysfunction'/de OR 'olfactory disorder'

The collected search terms were framed with Boolean Operators (OR and AND), tested for preliminary search across the individual databases and modified when there were unacceptable differences in search results between reviewers. For the database search, the search terms were separately developed for the particular databases.

For the PICO search, the four keywords were applied.

"Population" -Patients who were infected with severe acute respiratory syndrome coronavirus 2 (SARS-CoV-2) including new variants such as B.1.1.529, Alpha (B.1.1.7), Beta (B.1.351), Gamma (P.1), Delta variant, Omicron variant, Omicron "stealth" variant (BA.1, BA.2).

"Interventions" -Oral corticosteroid, Intranasal corticosteroid, Systemic corticosteroid, Intranasal sodium citrate, Smell/Taste/Olfactory training, Essential oil, Diets, Supplements,

Intranasal Vitamin A, Systemic omega-3, Coffee, Lavender syrup, Vaccination, Inflawell syrup, Carbamazepine, Palmitoylethanolamide (PEA) and luteolin

"Comparison" -Between treatments/interventions and placebo treatments for COVID-19-related olfactory dysfunction, clinical improvements of COVID-19-related olfactory dysfunction following treatments/ interventions

"Outcomes" -Time to clinical improvements from COVID-19-related olfactory dysfunction especially loss of smell following treatment interventions

**Developing forms.** The researcher developed the inter-reviewer agreement form and data extraction template by using Excel sheets. To ensure that these forms were pilot-tested, revised and modified. The operational definitions of all variables and items inserted in the developed forms were mentioned (Table 2) and shared with the co-reviewer and supervisor.

## Details of selection process

The reviewers followed the flow of the PRISMA 2020 selection frame, in which the reviewers had to be completed three steps: identification, screening and selection. Firstly, the reviewers identified if there was a previous similar review and counted the number of previous similar reviews. Besides, the reviewers used the predetermined core search terms for identifying the titles, abstracts, citations and reference lists. Among search results that appeared via medical databases, the reviewers noted the number of records identified for each search. In counting, the reviewers ensured the recording of the duplicated numbers among medical databases and searches. Secondly, the reviewers applied the inter-reviewer agreement form to screen all includable and downloaded studies if they are selected or not selected or may be selected. Here, the reviewers ensured that the reasons for the excludable studies were noted. Thirdly, the reviewers repetitively read the final selected papers, discussed among co-reviewer and supervisor and took agreements.

## Data extraction and analysis

The principal reviewer gathered the information of interest including basic characteristics by coding the relevant data items and putting them into the final version of the data extraction template. The co-reviewer double-checked the extracted data items with the original themes of the primary studies. Types and effects of treatments or interventions that were structured in the data extraction template were the open fields, from which the additional items of interest could be added. The data analysis had two portions. In the first portion, the researcher summarized the extracted items regarding the characteristics of the included studies, aggregated the summary data using frequencies and percentages and descriptively presented all aggregated information. In the second portion, the main interventions and their effects on the duration of COVID-19-related olfactory dysfunction were narratively analyzed and displayed in the individual paragraph.

## Critical appraisal tools

JBI critical appraisal tools were used for assessing the reporting qualities of the RCTs (Randomized Controlled Clinical Trials) and quasi-experimental studies selected. Especially, these tools were applied to assess the abilities of the included studies to reduce the biases in their designing process, research procedures and analysis process. The process of critique was completed by two independent appraisers and the overall appraisal of each included study was also demonstrated according to the agreed results of the particular item in the checklist in terms of "high" (agreed with a particular item), "low" (not agreed with a particular item) and "unclear"

**Table 2. Operational definitions of the variables in the inter-reviewer agreement form and data extraction template.**

| Variables | Operational Definition | Example |
|---|---|---|
| **Inter-reviewer Agreement Form** | | |
| Paper title | Full name of the selected paper title as published in the particular journal. It can appear on the first page or title paper of the published paper | Corticosteroid nasal spray for recovery of smell sensation in patients with COVID-19: A randomized controlled trial |
| Authors | Author's surname and year (in brackets) When there are three or fewer authors, all authors are mentioned. When four and more authors are included, only the first author is mentioned and followed by 'et al.' | Rashid RA, Zgair A and, Al-Ani RM (2021) Fiani B. et al., (2020) |
| Paper code | Code number of the full-text papers downloaded for screening | 2020-PubMed-0001 |
| Search Engine | Name of information source (Database) that is applied for a comprehensive search | EMBASE and PubMed |
| Yes | Note as "Yes" when the eligible paper is selected for the review's analysis | Yes |
| No | Note as "No" when the eligible paper is rejected for the review's analysis | No |
| Maybe | Note as "Maybe" when the reviewer's decision is uncertain for accepting or rejecting | Maybe |
| Reason | If "No" or "Maybe", give un-includable reason(s) | Intervention is not related to the COVID-19-induced olfactory dysfunction |
| **Data Extraction Template** | | |
| Paper title | Full name of the selected paper title as published in the particular journal. It can appear on the first page or title paper of the published paper | Corticosteroid nasal spray for recovery of smell sensation in patients with COVID-19: A randomized controlled trial |
| Authors | Author's surname and year (in brackets) When there are three or fewer authors, all authors are mentioned. When four and more authors are included, only the first author is mentioned and followed by 'et al.' | Rashid RA, Zgair A and, Al-Ani RM (2021) Fiani B. et al., (2020) |
| Publication Year | Only the year of publication is to be mentioned, not including day and month | 2020 |
| Country of Study | Name(s) of the country(ies) where the research is carried out | French |
| Study Objective (s) | Specific objective(s) of the studies screened for the review's analysis. All texts of the study objective(s) mentioned in the original paper need to be paraphrased. | To discover the effect of coffee on the taste and smell dysfunction following COVID-19 |
| Study Design | Design of the study(ies) downloaded for screening and as written in the methodology section | Randomized Control Trial |
| Sampling Method | Method(s) of sample selection as written in the methodology section | Block randomization |
| Study Population | Types and characteristics of the sample selected from the target population | Outpatients with COVID-19 anosmia |
| Sample size | Number of the sample that is defined for the study analysis | 118280 |
| Intervention(s) | Identification of treatment options for the COVID-19-related loss of smell | Systemic corticosteroid |
| Effects of intervention(s) | Number of days or hours between the confirmed date of the COVID-19-related loss of smell and date of recovery after treatment intervention | 5 hours or 5 Days |
| Study limitation (s) | All contents of the study limitations mentioned in the original papers need to be paraphrased | Self-reported olfactory and gustatory loss, aimed only at symptomatic subjects, lack of analysis of the differences between nonimmunized and immunized subjects, no information regarding the virus strain that infected the patients |
| Quality of study report | As the results of the report quality of the selected papers after being appraised by the JBI critical appraisal tool | Could appropriately inform the synthesis and interpretation of the study outputs |

(missed or un-reported) to inform the review's analysis and interpretation of the extracted data. Here, the review did not try to exclude the studies with many low or unclear results.

## Results

### Details of selection process

In the identification stage, the principal reviewer identified 1568 (PubMed), 1329 (PubMed Central) and 1401 (EMBASE) titles and abstracts while the second reviewer identified 1449 (PubMed), 1120 (PubMed Central) and 1501 (EMBASE) titles and abstracts. Additionally, the principal and second reviewers identified 198 titles and abstracts, and 201 titles and abstracts were identified respectively through the continuous identifications of the citations and references. Accordingly, a total of 8767 records could be identified initially. Here, 7056 records were determined by Endnote software as duplications and therefore 1711 records were for the screening stage after removing the duplicated records. In the screening stage, 1690 records were removed again due to many reasons for ineligibility for providing this review question's destinations and objectives (S1 Fig). In the selection stage, only 21 (10 Randomized Control Trials + 11 Quasi-experimental Studies) [36–56] were eligible for this review's inclusion and exclusion criteria and selected for further analysis.

### Characteristics of the included studies

Of the 21 intervention studies that matched the predetermined criteria for inclusion, one RCT was published in 2020 (n = 1, 4.8%), seven RCTs and five QES in 2021 (n = 12, 57.1%) and two RCTs and six QES in 2022 (n = 8, 38.1%) respectively. Regarding the country of origin where the studies were undertaken, four studies (19.1%) took place in Iran, two studies (9.5%) in Italy, two studies (9.5%) in Turkey, and two studies (9.5%) in Egypt. The rest 52.4% were carried out in French, Belgium, Iraq, Saudi Arabia, European Multicenter, India, Germany, United Kingdom, USA, Brazil and Europe and composed of an equal proportion of 4.76%. Regarding the study design, two designs; randomized control trial and a quasi-experimental design were included, in which 10 RCTs (47.6%) [36–38, 40, 44–49] and 11 QES (52.4%) [39, 41–43, 50–56] were counted. The median age of the 130,550-study population included in this review was 37 years (ranging from 24 to 69 years) and the median sample population per study was 100 (ranging from 10 to 118280). Other details were displayed in Table 3.

**Reporting quality of the included studies.**   The results of the reporting qualities of the RCTs and quasi-experimental studies were separately demonstrated in Tables 4 and 5 Among RCTs, the quality statuses of some studies were mentioned as "low" and "unclear" because their reports missed describing allocation concealment, blinding of participants, treatment providers and analysts, completion of follow-up, intention-to-treat (ITT) analysis, reliability of outcome measurements and appropriateness of research conduction and analysis. Also, among quasi-experimental studies, some were analyzed as "low" because their reports did not include control groups, completion of follow-up, consistent measurements and desirable outcomes.

**Interventional treatments for olfactory dysfunction.**   Steroidal intervention.
Generally, two types of corticosteroids were identified concerning corticosteroids administered for the treatment of COVID-19-related anosmia, oral and nasal. Nasal corticosteroids used were fluticasone nasal spray, betamethasone nasal drop, mometasone furoate nasal spray, and triamcinolone nasal saline irrigation (Table 6).

**Mometasone furoate nasal spray.**   In the trial of Abdelalim et al., 2021 [36], mometasone furoate nasal spray was administered 2 puffs (100 ˚g) once daily in each nostril for 3 weeks to 50 subjects in the study group along with the olfactory training which was instructed to perform smelling of different scents of rose, lemon, and clove for 20 seconds twice daily. The control group (50 subjects) was instructed to perform olfactory training alone for 3 weeks. The

**Table 3. Characteristics of the included studies.**

| Specifications | Frequency | Percentage |
|---|---|---|
| **Country of Origin** | | |
| Iran | 4 | 19.0% |
| Egypt | 2 | 9.5% |
| Europe | 2 | 9.5% |
| Italy | 2 | 9.5% |
| Turkey | 2 | 9.5% |
| Belgium | 1 | 4.8% |
| Brazil | 1 | 4.8% |
| French | 1 | 4.8% |
| Germany | 1 | 4.8% |
| India | 1 | 4.8% |
| Iraq | 1 | 4.8% |
| Saudi Arabia | 1 | 4.8% |
| UK | 1 | 4.8% |
| USA | 1 | 4.8% |
| **Focus Fields** | | |
| Knowing the effectiveness of different forms of corticosteroid medications for olfactory dysfunction following COVID-19 | 7 | 33.3% |
| Studying the effectiveness of different forms of non-corticosteroid medications for olfactory dysfunction following COVID-19 | 5 | 24.8% |
| Determining the effects of COVID-19 vaccination | 3 | 14.3% |
| Discovering the effect of non-medicated things on the taste and smell dysfunction following COVID-19 | 2 | 9.5% |
| Determining the effect of olfactory training (visual stimulation) in patients with COVID-19 anosmia for more than one month | 2 | 9.5% |
| Determining the effectiveness of various intervention methods on post-COVID-19 parosmia | 1 | 4.8% |
| Assessing the desire of the patients with post-infectious olfactory dysfunction to participate in research | 1 | 4.8% |
| **Study Designs** | | |
| Quasi-experimental Study (QES) | 11 | 52.4% |
| Randomized Control Trial (RCT) | 10 | 47.6% |
| **Data Collection Methods & Tools** | | |
| Sniffin Sticks Test, TDI (Transition Dyspnea Index) scores | 5 | 24.8% |
| 11-point visual analogue scale (VAS) | 4 | 19.0% |
| Duration for smell recovery | 3 | 14.3% |
| Online questionnaire | 1 | 4.8% |
| 11-point visual analogue scale (VAS) and (UPSIT)- (University of Pennsylvania Smell Identification Test) Test | 1 | 4.8% |
| Modified Arabic 20- Items Sino-Nasal Outcome Test (MA-SNOT-20) | 1 | 4.8% |
| Subjective smell assessment by using 5 different smells | 1 | 4.8% |
| Self-Rating Olfactory Score (SROS), and Olfactory Dysfunction Duration (ODD) | 1 | 4.8% |
| Physical examination and Lab investigation | 1 | 4.8% |
| Ad hoc semi-structured questionnaire | 1 | 4.8% |
| Single-item 5-point Likert question | 1 | 4.8% |
| Magnetic Resonance Immunosorbent Assay, questioning each patient | 1 | 4.8% |
| **Intervention** | | |
| Steroidal nasal drop/spray | 5 | 23.8% |
| Olfactory Training | 3 | 14.3% |
| Fully vaccinated with the COVID-19 vaccine | 3 | 14.3% |
| Consumption of 15–30 mg of Coffee | 1 | 4.8% |
| Administration of lavender syrup | 1 | 4.8% |
| Oral corticosteroid | 1 | 4.8% |
| Zinc supplement | 1 | 4.8% |

*(Continued)*

**Table 3.** (Continued)

| Specifications | Frequency | Percentage |
|---|---|---|
| Intranasal plus oral steroid treatment | 1 | 4.8% |
| Oral palmitoylethanolamide plus luteolin | 1 | 4.8% |
| Oral corticosteroid (OC) plus olfactory training (OT) | 1 | 4.8% |
| Inflawell syrup | 1 | 4.8% |
| Omega-3 fatty acid | 1 | 4.8% |
| Oral carbamazepine | 1 | 4.8% |
| **Publication Year** | | |
| 2021 | 12 | 57.1% |
| 2022 | 8 | 38.1% |
| 2020 | 1 | 4.8% |

recovery of smell function was subjectively measured at the end of each week with a visual analogue scale (VAS) from 0 to 10 answered by the patients themselves. The average time of recovery of olfactory dysfunction in the study group and the control group was 26.41 ± 7.99 days and 26.15 ± 5.07 days respectively with a p-value of 0.88. It was visible that there was no significant difference between the two groups.

**Mometasone furoate nasal spray with olfactory training.** In another randomized double-blinded controlled trial design, the treatment group was given mometasone furoate nasal spray 2 puff (100 ˚g) twice daily in each nostril for 4 weeks combined with olfactory training while the control group was given saline nasal spray instead of the corticosteroid with an identical treatment setting. Among the subjects of the treatment group, 19 completely recovered their sense of smell and 19 others became mild microsmic after 4 weeks of intervention by the time the research endpoint was reached while 8 fully recovered and 11 became mild microsmic within the control group. The VAS scores and Iran SIT scores of the patients in both groups did not significantly improve, the clinical recovery rate between the two groups significantly differed showing better improvement within the subjects of the intervention group (97.4% and 50% respectively [37].

**Betamethasone nasal drop.** Rashid et al. reported the subjects in the treatment group (138 subjects) were made to take betamethasone nasal drop 3 drops in each nostril thrice daily at the Kaiteki positions until recovery for about one month whereas the control group (138 subjects) was given 0.9% NaCl (Sodium Chloride) drops as placebo, with the frequency and the duration being the same as the former group. The median recovery time (7 days) of COVID-19-related anosmia was no significant difference between the intervention group and the placebo group [38].

**Fluticasone nasal spray.** In the trial of Singh, Jain and Parveen, 2021, the 60 subjects in the treatment group were treated for their anosmia with fluticasone nasal spray 2 puffs OD for 5 days and those (60 subjects) in the control group were not administered any form of steroid. The measurement tool used was subjective testing of smell by using 5 different strong odours. At the endpoint of the study, (88.33%) in the treatment group recovered their sense of smell completely while the olfactory function of the participants in the control group worsened [39].

**Nasal triamcinolone acetonide.** In Yildiz et al.'s study, a total of 150 patients with COVID-19 anosmia were randomly assigned to three groups with three different treatment options. The first group was not provided with any additional treatment for anosmia, the second group participants were treated with nasal irrigation with normal saline, and the subjects of the third one were administered nasal saline irrigation with triamcinolone acetonide. In that study, the mean recovery period of the participants from the third group was 5.6 ± 3.2

**Table 4. Reporting quality of RCT studies included in the review.**

| Studies | True Randomization | Allocation Concealment | Similarity between groups | Blinding of participants | Blinding of treatment provider | Blinding of outcome assessor | Equal provision of care | Completion of follow up | ITT analysis | Same measurement | Reliability of outcome measurement | Appropriate analysis | Appropriateness of RCT design |
|---|---|---|---|---|---|---|---|---|---|---|---|---|---|
| Hosseini, Mirmahdi, and Moghaddam, 2020 [44] | High | Unclear | High | Unclear | Unclear | Unclear | Low | High | Unclear | High | Low | High | Low |
| Hashem-Dabaghian et al., 2021 [45] | High | Unclear | High | Unclear | Unclear | Unclear | High | High | Unclear | High | High | High | Low |
| Abdelalim et al., 2021 [36] | High | Unclear | High | Unclear | Unclear | Unclear | High | High | Unclear | High | High | High | Low |
| Rashid, Zgair, and Al-Ani, 2021 [38] | High | High | High | High | High | Unclear | High | High | Unclear | High | High | High | High |
| Kasiri et al., 2021 [37] | High | High | High | High | High | Unclear | High | Low | Unclear | High | High | High | High |
| Abdelmaksoud et al., 2021 [47] | High | Unclear | High | Unclear | Unclear | Unclear | High | High | Unclear | High | High | High | Low |
| D'Ascanio et al., 2021 [48] | High | Unclear | High | Unclear | Unclear | Unclear | High | Low | Unclear | High | High | High | Low |
| Yildiz et al., 2021 [40] | Low | Unclear | High | Unclear | Unclear | Unclear | High | High | Unclear | High | High | High | Low |
| Tond et al., 2022 [46] | High | High | High | High | High | Unclear | High | Low | Unclear | High | High | High | High |
| Hernandez et al., 2022 [49] | High | Low | High | Low | Low | Unclear | High | Low | Unclear | High | High | High | Low |

days while the subjects in other groups showed the duration of olfactory dysfunction as being nearly twice longer than that of the former [40].

**Nasal and oral corticosteroid.** The result of a quasi-experimental study showed no significant difference in olfactory dysfunction recovery rates between the treatment with nasal and oral corticosteroid groups and no treatment group. Moreover, most of the patients in the study did not recover their sense of smell till the end of the study regardless of taking treatments [41].

**Oral or nasal corticosteroid with olfactory training.** A quasi-experimental type of multi-center study compared the effectiveness of oral corticosteroids with olfactory training (group 1), a nasal corticosteroid with olfactory training (group 2), and olfactory training alone (group 3) on COVID-19-related anosmia and hyposmia. Though there was a significant difference between the three groups after one month from baseline, no significant difference between them after two months from baseline [42]. A Belgium study's result showed a significant difference between the changes in olfactory scores (TDI scores-Threshold, Discrimination, Identification scores) of the persistent dysosmia patients of the two groups. The subjects in the intervention group (Oral corticosteroid 10-day course plus olfactory training) significantly

**Table 5. Reporting quality of quasi-experimental studies included in the review.**

| Studies | Clarity of cause and effect | Similarity between groups | Equal provision of care | Inclusion of control group | Multiple measurements of outcomes | Completion of follow up | Same measurement | Reliability of outcome measurement | Appropriate analysis |
|---|---|---|---|---|---|---|---|---|---|
| Denis et al., 2021[55] | High | Low | Low | Low | Low | High | High | High | High |
| Bon et al., 2021[43] | High | High | High | High | High | Low | High | High | High |
| Khalifah et al., 2021 [41] | High | High | High | High | High | High | High | High | High |
| Saussez et al., 2021 [42] | High | High | High | High | High | High | High | High | High |
| Singh, Jain, and Parveen, 2021 [39] | High | High | High | High | High | High | High | High | High |
| Altundag, Yilmaz, and Kesimli, 2022 [56] | High | High | High | High | High | Low | High | Low | High |
| Porru et al., 2022 [51] | High | High | High | High | High | High | High | High | High |
| Pendolino et al., 2022 [54] | High | Low | Low | Low | High | Low | Low | Low | High |
| Wisnivesky et al., 2022 [53] | High | High | High | High | High | Low | High | Low | High |
| Vasconcelos et al., 2022 [50] | High | Low | Low | Low | High | High | Low | High | High |
| Vaira et al., 2022 [52] | High | High | High | High | High | Low | High | Low | High |

improved their TDI scores after 10 weeks although they were not still normosmic compared to the control group (olfactory training alone) [43].

## Non-steroidal supplementary intervention

**Coffee.** The quasi-experimental study on 118341 subjects was divided into intervention (93915) and control (22426) groups, those who took coffee and did not consume coffee. And the subjects were subdivided into (5) sub-groups according to the absence and presence of two or more of the following underlying diseases: diabetes, hypertension, and heart disease. In the intervention group, one sub-group without underlying disease was given 15–20 mg of coffee and four sub-groups with underlying diseases were given 20–30 mg of coffee. It was investigated that coffee consumption had an influence on recovering patients since (98.3%) in the coffee group but only (1.3%) recovered from the smell and taste dysfunction. The average recovery time of the patients with the absence of underlying disease was 5-7h and that of the patients with the presence of underlying diseases was (2–4 days) [44].

**Table 6. Interventions and their effects on the COVID-19-related olfactory dysfunction.**

| Study | Type of Intervention | Effects of Intervention (Average duration of smell loss) |
|---|---|---|
| Abdelalim et al., 2021 [36] | Mometasone furoate nasal spray 2 puff once daily for 3 weeks along with olfactory training | approximately 26 days in the treatment group and control group (P = 0.88) |
| Kasiri et al., 2021 [37] | Nasal corticosteroid, mometasone furoate was given 2 puff BD (Twice a day) for 4 weeks + olfactory training | 4 weeks (97.4%) in the treatment group 4 weeks (50%) in the control group (P < 0.05) |
| Rashid, Zgair, and Al-Ani, 2021 [38] | Betamethasone nasal drop 3 drops into each nasal cavity three times daily for one month | 7 days (84%) in the control group 7 days (82%) in the treatment group (P = 0.87) |
| Singh, Jain, and Parveen, 2021 [39] | Fluticasone nasal spray 2 puffs OD (Once a day) for 5 days | Within 5 days (88.33%) in the treatment group Within 5 days (0%) in the control group (P < 0.001) |
| Yildiz et al., 2021 [40] | Nasal saline irrigation plus triamcinolone acetonide nasal spray | 5.6 +/- 3.2 days in the treatment group (100%) 11.2 +/- 6.4 days in the control group (P < 0.001) |
| Khalifah et al., 2021 [41] | Intranasal plus oral steroid treatment | No significant effects during one month to the longest being about 7 months in both groups (P = 0.67) |
| Saussez et al., 2021 [42] | Oral corticosteroid (OC) plus olfactory training (OT) | one month (75%) in the treatment group Later recovery time in the control group (P = 0.35) |
| Le Bon et al., 2021 [43] | Oral corticosteroid (methylprednisolone 32 mg) once daily for 10 days along with instructions to perform olfactory training for 10 weeks | An average increase in TDI scores of 7.7 points in the treatment group and 2.1 points in the control group (P < 0.05) |
| Hosseini, Mirmahdi, and Moghaddam, 2020 [44] | 15–20 mg of coffee for the group without underlying diseases 20–30 mg of coffee for the group with underlying diseases | 5–7 hours (Patients without underlying diseases) 2–4 days (Patients with underlying diseases) (P < 0.05) |
| Hashem-Dabaghian et al., 2021 [45] | 9ml of lavender syrup two times a day for 3 weeks | significant improvement in VAS score among (91.3%) of the treatment group after 3 weeks (P < 0.05) |
| Barzin Tond et al., 2022 [46] | Inflawell syrup 10 ml thrice daily for 14 days | 14 days in the treatment group Later recovery time in the control group (P< 0.05) |
| Abdelmaksoud et al., 2021 [47] | Zinc supplement (zinc sulfate 220 mg which is equivalent to 50 mg of elemental zinc) twice daily along with the standard treatment of Egyptian treatment guideline | 7 days (5–9 days) in the treatment group 12 days (8–17 days) in the control group (P < 0.05) |
| D'Ascanio et al., 2021 [48] | Oral supplementation with palmitoylethanolamide 700 mg plus luteolin 70 mg once daily for 30 days + olfactory training | TDI scores of the two groups did not significantly differ (P = 0.67) |
| Hernandez et al., 2022 [49] | Omega-3 fatty acid 2 capsules twice daily for 12 weeks with olfactory training | 12 weeks (27.6%) in the treatment group (20.7%) in the control group (P = 0.05) |
| Vasconcelos et al., 2022 [50] | Carbamazepine for 11 weeks; the dosage starting with 100 mg daily for 3 consecutive days and titrating gradually by increasing the dose to 100 mg per day for 5 consecutive days to reach a maximum of 400 mg per day until signs of improvement | 51.5 days (range = 36–118) (90%) in the treatment group (No control group) (P < 0.001) |
| Porru et al., 2022 [51] | Fully vaccinated with the COVID-19 vaccine | 1–3 days in the fully immunized group (P < 0.001) |
| Vaira et al., 2022 [52] | Fully vaccinated with the COVID-19 vaccine | No significant difference in the intervention and control groups (P = 0.68) |

*(Continued)*

**Table 6.** (Continued)

| Study | Type of Intervention | Effects of Intervention (Average duration of smell loss) |
|---|---|---|
| Wisnivesky et al., 2022 [53] | Fully vaccinated with the COVID-19 vaccine | No significant difference in the intervention and control groups (P = 0.49) |
| Pendolino et al., 2022 [54] | Olfactory training for 6 months | 6 months (25.4%) in the intervention group (No control group) |
| Denis et al., 2021 [55] | Olfactory Training (maximum of 16 weeks) | 28 days in the intervention group (No control group) |
| Altundag, Yilmaz, and Kesimli, 2022 [56] | Modified olfactory training (MOT) for 9 months | Within 9 months, more improvement in TDI scores in the intervention group and less improvement in the control group (P < 0.05) |

**Lavender syrup.** To test the effect of lavender syrup on COVID-19-induced olfactory dysfunction, 43 patients with olfactory dysfunction were divided by block randomization into two groups, the lavender group (23) and the control group (20). The participants from both groups were treated according to the standard guidelines of the Iran Ministry of Health. The lavender group participants were additionally given 9ml of lavender syrup twice daily for 3 weeks. After assessment with the Visual Analog Scale (VAS), the result of the study showed significant improvement in olfactory function after 3 weeks of intervention in the lavender group than in the control group [45].

**Inflawell syrup.** Another study which was also conducted in Iran tested the effectiveness of another syrup called inflawell syrup in minimizing disease progress of COVID-19-related olfactory dysfunction. A total of 47 eligible patients were recruited and divided into two groups: 24 subjects in the intervention group who received 10 ml of inflawell syrup three times a day for 14 days and 23 subjects in the control group who were given a placebo syrup of similar taste and similar colour to those of inflawell. Since the assessment points for the study were just two points: at the baseline (Day 0) and the endpoint (Day 14), 19 from the intervention group and 20 from the control group were of normal olfactory function at the baseline of the study and therefore, the actual duration of recovery of the sense of smell cannot be explored. After 14 days, the 5 subjects with weak olfactory function from the intervention group fully recovered while 3 subjects from the control group were indifferent to their situation at the baseline of the study. The median duration of smell recovery was not mentioned in detail [46].

**Zinc.** In a randomized control trial in Egypt, a total of 105 patients with COVID-19 anosmia was randomly assigned to two groups: the intervention group (49) received a zinc supplement twice daily in the form of zinc sulfate 220 mg which was equivalent to elemental zinc 50 mg) and the control group (56) who were not administered zinc supplement. The follow-up period was until the full recovery of both the olfactory dysfunction and the other symptoms of the disease. The median duration of recovery in the intervention group (7 days (5–9 days)) was significantly different (P<0.05) from that in the control group (18 days (14–22 days)) [47].

**PEA and luteolin.** In another study by (D'Ascanio et al., 2021), to examine the roles of the neuroprotective agents with anti-inflammatory properties in treating COVID-19-related olfactory dysfunction, a total of 12 patients with COVID-19related olfactory dysfunction were randomly assigned into two groups. While the control group (5 participants) had to perform olfactory training alone, 7 subjects in the intervention group were daily administered oral palmitoylethanolamide (PEA) 700 mg plus luteolin 70 mg as an adjuvant to olfactory training. The results were assessed at the baseline (Day 0) and the endpoint at 30 days by using 0 to 48 scales Sniffin' Sticks Test with TDI scores. The TDI scores of the intervention group

significantly improved while the TDI scores in the control group did not differ from the baseline. The exact mean duration of recovery for the treatment group participants could not be explored because of short-term follow-up [48].

**Omega-3.** In a randomized control trial, 58 patients with post-viral anosmia were randomly assigned into two groups: 29 subjects in the treatment group who receive omega-3 fatty acid (anti-inflammatory and neuro-regenerative supplement) 2 capsules twice daily for 12 weeks and the control group (29 subjects) who were directed to perform olfactory training alone for 12 weeks. There were two assessment points in the whole study, at the start and the end with 0 to 48 scales Sniffin' Sticks Test with TDI scores. Only (27.6%) of the intervention group and (20.7%) of the control group became normosmic during the study period. Only the threshold scores of the treatment group significantly improved during the study period [49].

**Carbamazepine.** In a study by Vasconcelos et al., (2022), patients with COVID-19 anosmia were treated with carbamazepine (an anticonvulsant drug) for a maximum duration of 11 weeks without a control group. The dose was initially given as 100 mg per day for 3 consecutive days, then gradually titrated by increasing the dose from 100 mg every 5 days to a maximum dose of 400 mg/ day. After evaluating the improvement by questioning, 90% of patients responded with improvement. The median duration of recovery was 51.5 days (Range = 36–118) [50].

**COVID-19 vaccination.** In a quasi-experimental study by Porru et al., (2022) testing the effect of the COVID-19 vaccination on COVID-19-related anosmia, a total of 165 health workers from Verona Hospital, Italy was categorized as unvaccinated (114), partially vaccinated (17), and fully immunized (34). Regarding anosmia, 45% of unimmunized experienced anosmia whereas 6% of partially immunized and 18% of fully immunized did ($P<0.001$). The author mentioned that vaccination seems to be effective regarding the duration of symptoms (1–3 days, 4-7days, and > 7 days) [51]. In contrast, a quasi-experimental study by (Vaira et al., 2022) found that there was no significant difference in the prevalence of anosmia between the fully vaccinated and the unvaccinated subjects [52]. Moreover, another quasi-experimental study (which the author described as a prospective cohort study) conducted in New York City showed no significant difference between the two groups (vaccinated and unvaccinated) for anosmia from the baseline to the 6-month follow-up period [53].

**Olfactory training.** In a quasi-experimental study conducted in the United Kingdom, the effectiveness of olfactory training on post-infectious olfactory dysfunction was observed. Among 59 participants of the study, (25.4%) responded to the improvement in their smell function treated with olfactory training (OT) for 6 months. However, in that study, (45.8%) of the whole participants took other medications together with olfactory training [54]. Another quasi-experimental study without a control group also explored the effectiveness of olfactory training with the help of visual stimuli. After assessing with a VAS score of 0–10, the author concluded that olfactory training with visual stimuli aids significantly improved olfactory function in patients with COVID-19-related anosmia if the duration of training lasted for at least 28 days [55].

Sniffin' Sticks test with TDI scores (0–48) assessment tool was used to test the effect of long olfactory training. The result showed that, at the end of the study, the mean TDI scores of the patients in the treatment group (9 months modified olfactory training-MOT) improved more significantly than those in the control group (no MOT). Although not all the patients in the intervention group recovered during the study period, the mean TDI score after 9 months of MOT (27.9) was close to the normosmic threshold (30.3) and therefore, the duration of smell recovery for the patients who adopt MOT was about 9 months [56].

## Discussion

This review specialized in the knowledge of the possible treatment options for olfactory disorders among patients with COVID-19. The main aim of this review was to make a comparison between the efficacy of corticosteroid treatment options that have emerged as the most commonly used treatment interventions among patients with COVID-19 olfactory loss across the globe, and the effectiveness of other medical or non-medical interventions. This review was different from the previous systematic reviews and meta-analyses in terms of the number of included studies and sample sizes, study population, the specific aetiology and direct effects of treatment strategies on the COVID-19 induced olfactory dysfunction.

This review analyzed and concluded ten medical and non-medical interventions; corticosteroids, zinc, carbamazepine, omega-3, palmitoylethanolamide (PEA) plus luteolin, vaccination, olfactory training, coffee, lavender syrup, and inflawell syrup. In this conclusion, when administering mometasone furoate nasal spray (2 puffs once daily along with olfactory training) to patients with COVID-19 olfactory dysfunction, the recovery time (more than 26 days) was the same between the treatment group and intervention group [36]. Contrariwise, a study [37] demonstrated that the group treated with mometasone furoate nasal spray 2 puffs (100 ˚g) twice daily and intervened with olfactory training for 4 weeks had a higher recovery rate (97.4%) than the control group (50%). Because of these conclusions, it is assumed that a daily single dose of mometasone furoate nasal spray has no significant effect on the COVID-19-related olfactory dysfunction, but its daily twice dose was efficacious. On the other side, it can be assumed that mometasone furoate in its form of nasal corticosteroid spray might not be effective against the olfactory dysfunction following COVID-19 infection, for which more advanced experimental studies should be conducted [37].

Besides, when the patients with COVID-19 were treated with betamethasone nasal drops (3 drops three times daily for one month) for their olfactory loss, the median recovery time of both intervention (82%) and control (84%) groups showed equally within 7 days [38]. From the point of this conclusion, betamethasone nasal drops could not be effective.

The treatment option of fluticasone nasal spray 2 puffs once daily showed a significant improvement of the olfactory dysfunction among the treatment group that had a recovery proportion (88.3%) on the fifth day of intervention whereas no recovery rate occurred among the control group [39]. Because of this favourable result, that review also concluded that the use of fluticasone nasal spray significantly improves olfactory function in patients with anosmia following COVID-19 infection.

The patients with COVID-19-related odour loss returned to their normal condition of olfactory function within 5.6 +/- 3.2 days after the treatment option of nasal saline irrigation and triamcinolone acetonide nasal spray while the control group had a double recovery time (11.2 +/- 6.4 days) [40]. Therefore, it can be concluded that this treatment option strongly improves the olfactory function of the patients with COVID-19-related anosmia.

However, the intervention group who took the combination of intranasal and oral corticosteroids showed their normal state of the olfactory function between 1–7 months and this duration was the same in the control group [41]. From this result, this review was difficult to determine the effectiveness of oral corticosteroids because their study sample was very small and the mean duration of recovery was not mentioned clearly in their paper. Considering another point, their report mentioned that more than 75% of the subjects in group one (oral corticosteroid plus olfactory training) recovered from the COVID-19-related smell loss about one month of intervention. Because of this fact, it can be assumed that oral corticosteroids might shorten the duration of olfactory dysfunction, but further experimental studies that can provide more accurate results should be carried out.

Further, a significant improvement in TDI scores occurred among patients with COVID-19 smell loss from the intervention group after 10 weeks of interventions of oral corticosteroid (methylprednisolone 32 mg once daily) and olfactory training [43]. However, this study did not completely describe the median recovery time and proportions in both the intervention and control groups. Generally, from their results, it can be concluded that the use of short-course oral corticosteroid therapy possibly benefits the patients with COVID-19-induced persistent dyssomnia.

In other treatments or interventions rather than corticosteroids, caffeine in coffee was a considerable treatment option for the COVID-19-associated olfactory dysfunction. In the study [44], the patients with COVID-19 who were intervened with 15–30 mg of coffee had a full recovery from the COVID-19-related smell loss between a few hours and 4 days. This review could conclude from this intervention result that coffee consumption had an improved effect on the olfactory function. When comparing the effects of caffeine with the efficacy of corticosteroids on the smell loss following the COVID-19 infection, the shorter median recovery time was remarkable among the group with the intervention of caffeine. However, more trials on the effects of caffeine should be carried out [44].

Another commonly used treatment option for the COVID-19-induced smell loss was taking 9ml of lavender syrup twice daily. In this option, the vast majority (91.3%) of the patients with COVID-19 anosmia significantly improved their olfactory dysfunction according to the results of the VAS score after 3 weeks of intervention [45]. When comparing this treatment option with the corticosteroid treatments, the median recovery time of this treatment was longer than the results of treatment options of fluticasone and triamcinolone acetonide nasal spray and shorter than the results of a daily double dose of mometasone furoate nasal spray and oral corticosteroid. Here, it can be concluded that fluticasone and triamcinolone acetonide nasal spray might be more efficacious.

When looking at the effectiveness of another treatment option called inflawell syrup, many patients with COVID-19 who suffered from the related olfactory dysfunctions gained their normal smell conditions within 14 days [46]. Like the treatment option of lavender syrup, the longer recovery time was noted compared to the effects of fluticasone and triamcinolone acetonide nasal spray and the shorter recovery time was found compared with the former efficacy of mometasone furoate nasal spray and oral corticosteroid.

Considering another treatment option called zinc supplements, the administration of zinc supplements significantly improved olfactory function in patients with COVID-19-related anosmia. With this intervention, a significantly shorter recovery time (approximately 7 days) was noted among the treatment group [47]. When making a comparison between this intervention and corticosteroid treatments, fluticasone and triamcinolone acetonide nasal spray were more likely to reduce the duration of recovery than this intervention, but this intervention could be more effective in reducing the duration of the COVID-19-related smell loss than the treatment options of mometasone furoate nasal spray and oral corticosteroid. A systematic review concluded that zinc was mentioned as an ineffective intervention for post-viral olfactory dysfunction, but their review did not include the experiences of the patients with COVID-19 [57].

Considering another supplement called combined supplementation of oral palmitoylethanolamide (PEA) and luteolin, the evidence showed that this intervention had no significant effects in reducing the duration of smell loss following the COVID-19 infection [48]. Besides, the oral supplement of omega-3 could not show any effect on the reduction of smell loss duration among patients with COVID-19 until 12 weeks of intervention [49]. When evaluating the effects of carbamazepine treatment, this treatment option could end the symptoms of the COVID-19-associated olfactory loss after about 52 days of intervention [50]. Here, this review

could conclude that the treatment options of corticosteroids were more efficacious than the supplementations of palmitoylethanolamide (PEA) and luteolin, omega-3 and carbamazepine.

In considering the effect of the COVID-19 vaccination on the reduction of smell duration due to the COVID-19 infection, two studies [52, 52] concluded that the full vaccination had no any affect, but a study [51] incomparably concluded that the patients with COVID-19 recovered from the associated smell loss within 1–3 days because of full vaccination. Here, the efficacy of the COVID-19 vaccination on the smell loss due to the COVID-19 infection was not unclear and therefore further trials should be undertaken [51–53].

Another commonly used intervention for the COVID-19-related anosmia was the olfactory training. In conclusion of this review, the average recovery time of the COVID-19-associated anosmia was 28 days because of olfactory training and 9 months because of modified olfactory training [55, 56]. In reducing the duration of olfactory dysfunction following the COVID-19 infection, the olfactory training was less likely to be effective than the treatment options of corticosteroids, but the olfactory training should be combined with other treatment or intervention options. Similarly, the systematic review by Helman *et al.* (2022) recommended that olfactory training should be the first-line intervention for returning to the normal function of the olfactory lobe [58]. A systematic review and meta-analysis also discussed the beneficial effects of the olfactory training varied according to the underlying causes and extents of head injuries, but the olfactory training was the general and effective intervention in treating the symptom of olfactory dysfunction. Their meta-analysis reported that there were higher spontaneous recovery rates of the post-traumatic olfactory disorders among the patients who received olfactory training as compared with other short-term medications [59]. In another report, the patients with post-viral olfactory dysfunction who took regular olfactory training were about 1.7 times to 4.6 times more likely to improve the disorder conditions of olfactory function than those without taking olfactory training [31]. In the conclusion of another systematic review, the combined interventions of corticosteroids and modified olfactory training were more proficient in the management of the post-viral olfactory disorders and they received maximum therapeutic effects of the combined interventions within 24–36 weeks. Nonetheless, this review recommended that when a patient with the COVID-19-related olfactory dysfunction has a desire to take the treatment option of corticosteroid, he or she should be examined for the underlying risk factors, thoroughly consult with a medical professional about the potential risks of corticosteroids and measure the impacts of smell loss on the quality of his or her life [60].

## Conclusion

The life quality of about two-thirds of the patients with COVID-19 is affected by the related olfactory dysfunctions. This systematic review came to identify possible treatments and interventions for the olfactory dysfunction that was defined as the consequence of the COVID-19 infection and to provide a better understanding of the effects of corticosteroid treatment options that are the most commonly applied intervention strategy among coronavirus disease 2019 patients by comparing the efficacies of other treatments and interventions that are currently available. This review could conclude that corticosteroid treatment is effective in reducing the duration of the COVID-19-related smell loss and olfactory training, the basic, essential and effective intervention, should be used as a combination therapy.

## Strengths and limitations of study

In the PICO frame applied for formulating the research question, some participant descriptions were not very much specific due to the lack of severity level of patients whether they are

mild to moderate patients or severe patients and it could lead to doubtful assumptions. However, the research question itself did not denote a directional hypothesis in which the tendency to introduce confirmation bias unintentionally or intentionally might be higher. Thus, this systematic review study could be generally viewed as free of confirmation bias. In this secondary research, there was a team formed before the protocol approval and there were regular intervals of supervision before submission. Moreover, a research protocol was developed and standardized among research team members. These partly minimized the possibility of experimenter bias. In this systematic review, it was the limitation that the studies from other databases than Pubmed, Pubmed Central, and Embase were not searched for and not selected for review. Considering reporting bias, outcome reporting biases could not arise because all of the selected previous studies were included in the analysis and all the significant and insignificant findings of the studies were reported in the finding and discussion sections. However, language bias could arise due to the pre-set inclusion criteria in which studies with the English language were solely chosen. Here, the publication bias could not be revealed because this study was just a systematic literature review without meta-analysis and Egger's test or funnel plot test could not be exercised. In this study, the PRISMA checklist was properly applied in place within each section starting from the title section.

The articles without free access could not be purchased and the results of these articles were desperately omitted. The research question of this study also limited the type of study designs and therefore, other important information about the COVID-19-related olfactory dysfunction which might be discovered from other types of study designs might be unquestionably missed. Generally, most of the trials excluded pregnant mother, child patients and critical patients from the participants of research and, therefore, there would be limitations to generalising the findings of this study to these populations.

## Dissemination plan

The full dissertation paper will be submitted to the libraries of the University of Bedfordshire, UK and STIMU, Myanmar. A short report of this review will be posted to the public through popular social media: Facebook and Twitter. The key findings of this study will be distributed to the members of the medical profession in Myanmar through the annual medical conference organized and held by the Myanmar Medical Association (MMA). Finally, as an academic research paper, the major processes and key outputs of this systematic review will also be published in the PLOS One, a peer-reviewed open-access scientific journal.

## Administrative information

The utilization of unauthorized intellectual properties is refrained by the authors. The whole paper is free from copying or stealing the ideas or artworks of other scholars, authors, and professional co-workers. This was the student dissertation work of Dr Phyo Zin Win which has been undertaken for the award of Master of Science in Public Health from the University of Bedfordshire (UK). The co-reviewer and supervisor had the responsibilities for designing the review, undertaking all processes of the systematic review and revising the manuscript together with the principal investigator. Another co-author had the responsibilities for selecting the studies per protocol, extracting the relevant data, and reviewing and editing the manuscript.

## Supporting information

**S1 Fig. PRISMA 2020 flow diagram for new systematic reviews which included searches of databases and registers only.**
(TIF)

**S1 Table. PRISMA (Preferred Reporting Items for Systematic Review and Meta-analysis) 2020 checklist.**
(DOCX)

**S2 Table. Objectives and measurement tools of the included studies.**
(DOCX)

**S1 File.**
(XLSX)

## Acknowledgments

The author sincerely and humbly gives special thanks to the supervisor, co-reviewer, all lecturers from STIMU and the University of Bedfordshire for their professional and academic guidance in conducting this systematic review research during the proposal preparation and dissertation periods, and my aunt for her financial support aa well as my friend and my lovely girlfriend for their psychosocial support during the stressful days of the dissertation period.

## Author Contributions

**Conceptualization:** Phyo Zin Winn, Thein Hlaing, Kyaw Myo Tun, Seim Lei Lei.

**Data curation:** Phyo Zin Winn, Thein Hlaing, Seim Lei Lei.

**Formal analysis:** Phyo Zin Winn, Thein Hlaing, Kyaw Myo Tun, Seim Lei Lei.

**Investigation:** Phyo Zin Winn, Seim Lei Lei.

**Methodology:** Phyo Zin Winn, Thein Hlaing, Kyaw Myo Tun.

**Project administration:** Phyo Zin Winn, Thein Hlaing, Kyaw Myo Tun.

**Resources:** Phyo Zin Winn, Thein Hlaing, Kyaw Myo Tun, Seim Lei Lei.

**Software:** Phyo Zin Winn, Seim Lei Lei.

**Supervision:** Thein Hlaing, Kyaw Myo Tun.

**Validation:** Phyo Zin Winn, Thein Hlaing, Kyaw Myo Tun.

**Visualization:** Phyo Zin Winn, Thein Hlaing, Kyaw Myo Tun.

**Writing – original draft:** Phyo Zin Winn, Thein Hlaing.

**Writing – review & editing:** Phyo Zin Winn, Thein Hlaing, Kyaw Myo Tun, Seim Lei Lei.

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
