## [Decision Letter · Decision Letter 0]

15 Nov 2022

PONE-D-22-27243Effect of any form of steroids in comparison with that of other medications on the duration of olfactory dysfunction in covid-19 patients: a systematic review of randomized trials and quasi-experimental studiesPLOS ONE

Dear Dr. Winn,

Thank you for submitting your manuscript to PLOS ONE. After careful consideration, we feel that it has merit but does not fully meet PLOS ONE’s publication criteria as it currently stands. Therefore, we invite you to submit a revised version of the manuscript that addresses the points raised during the review process.

We look forward to receiving your revised manuscript.

Kind regards,

Dong Keon Yon, MD, FACAAI

Academic Editor

PLOS ONE

Journal Requirements:

2.In your Data Availability statement, you have not specified where the minimal data set underlying the results described in your manuscript can be found. PLOS defines a study's minimal data set as the underlying data used to reach the conclusions drawn in the manuscript and any additional data required to replicate the reported study findings in their entirety. All PLOS journals require that the minimal data set be made fully available. For more information about our data policy, please see http://journals.plos.org/plosone/s/data-availability.

4. We note that Figure 2 in your submission contain [map/satellite] images which may be copyrighted. All PLOS content is published under the Creative Commons Attribution License (CC BY 4.0), which means that the manuscript, images, and Supporting Information files will be freely available online, and any third party is permitted to access, download, copy, distribute, and use these materials in any way, even commercially, with proper attribution. For these reasons, we cannot publish previously copyrighted maps or satellite images created using proprietary data, such as Google software (Google Maps, Street View, and Earth). For more information, see our copyright guidelines: http://journals.plos.org/plosone/s/licenses-and-copyright.

Additional Editor Comments:

Thank you for submitting your manuscript. The reviewers and I believe it is of potential value for our readers. However, the reviewers have raised a number of very important issues, and their excellent comments will need to be adequately addressed in a revision before the acceptability of your manuscript for publication in the Journal can be determined. We cannot guarantee that your revised paper will be chosen for publication; this would be solely based on how satisfactorily you have addressed the reviewer comments.

#1. Please delete Ref 1 and 7, which is not peer-reviewed article.

#2. World Health Organization (WHO) first declared the coronavirus outbreak as a global health emergency on 30 January 2020 -> Please cite the article such as DOI: https://doi.org/10.54724/lc.2022.e10

#3. Please change Ref 35 to 2020 PRISMA guideline (DOI: https://doi.org/10.54724/lc.2022.e9).

Reviewers' comments:

Reviewer's Responses to Questions

**Comments to the Author**

1. Is the manuscript technically sound, and do the data support the conclusions?

Reviewer #1: Yes

Reviewer #2: Yes

2. Has the statistical analysis been performed appropriately and rigorously? 

Reviewer #1: No

Reviewer #2: N/A

3. Have the authors made all data underlying the findings in their manuscript fully available?

Reviewer #1: Yes

Reviewer #2: Yes

4. Is the manuscript presented in an intelligible fashion and written in standard English?

Reviewer #1: No

Reviewer #2: Yes

5. Review Comments to the Author

Reviewer #1: • Abstract: JBI?

• Abstract: Why is it that the authors have recommended that the potential risks of steroids should be carefully considered? What findings from their review have led to this recommendation?

• Introduction-Line 76-current corona virus: Be more specific.

• Methods: PICOT?

• Line 140: Delete “Surely,”.

• Line 150: Is “abbreviation” a type of publication?

• Line 163: Pregnancy? Pregnant?

• Describe/define “mild”, “moderate” and “severe” COVID-19 patients either in the ‘introduction’ section or preferably in the ‘methods’ section as it had a bearing in the inclusion/exclusion criteria formulated by the authors of the present review.

• Line 189: When was the search performed?

• Line 199: PICO?

• Table 2 and Line 189: PubMed, PubMed Central and EMBASE are mentioned (Line 189). MEDLINE features in Table 2.

• Line 259: Research? Review?

• Lines 263-270, Ethical consideration: Not applicable; Delete.

• Lines 275-277: Why should there be a difference in the number of items identified by two different researchers/reviewers when the same search string is applied at the same database on the same day? This is not acceptable and points towards lapses in the search strategy. Screening of the identified items is done by 2 reviewers and discrepancies sorted out in consultation with a third reviewer. Identification of items does not require 2 reviewers. Isn’t it? And why is there a discrepancy in the numbers related to identification of items?

• Line 280: The total of “8767” records identified is wrong. Why are the numbers from PubMed added twice? The same is the case for PubMed Central and EMBASE. This is not appropriate.

• The quality of the figure related to the search strategy (PRISMA flowchart) is not good and should be replaced with a better-quality figure/picture. For related details, refer to the instructions to authors on the journal website.

• PRISMA flowchart: What is the difference between “Studies included in the review n=21” and “Reports of included studies n=21”?

• PRISMA flowchart: Registers (n=8767)? Registers (n=0)?

• PRISMA flowchart: The number of items/records identified is incorrect for reasons mentioned before (comment related to Lines 275-277).

• PRISMA flowchart: Records sought for retrieval (n=?)?

• PRISMA flowchart, Identification: The number of records/items identified from citation searching is not mentioned. Again, there are two different numbers mentioned for two different reviewers in the text (Line 278).

• S2 Figure is of poor quality and should be replaced with a better-quality one.

• Lines 284-285: Cite all the 10 RCTs.

• Lines 284-285: Cite all the 11 quasi-experimental studies.

• Line 289: Cite all the 21 studies included in the present systematic review.

• Line 354: “In the trial of (Singh, Jain and Parveen, 2021)”?

• Covid-19 vaccination: Why is this considered as a treatment strategy in the present systematic review? It shouldn’t be included. The related studies that the present authors have included have studied the occurrence of anosmia in COVID-19 vaccinated individuals infected with COVID-19.

• Line 503: Replace “Discussion on main findings” with “Discussion”.

• Line 509: Cite the previous systematic reviews and meta-analyses that are here referred to by the present authors. Besides, mention how was the present systematic review different in terms of the different parameters listed by the present authors. A Table depicting the list of related previous systematic reviews (plus the present systematic review) and the differences should be included. This is important to highlight the fact that the present systematic review is not redundant and the importance of publishing the present systematic review.

• Lines 603-609: There is a combination of Vancouver and Harvard referencing styles. Here and elsewhere, corrections should be made sticking to the Vancouver style.

• Lines 645-646: On what basis is it recommended that the potential risks of steroids should be carefully considered? What findings in the present systematic review have led the present authors to stating this recommendation?

• Line 669: Delete “desperately”.

• Lines 692-696 – Administrative information: Mention who is the student awarded the Master’s degree and the administrative roles of all others included as co-authors. Hasn’t the supervisor fulfilled the authorship criteria? The supervisor seems to be acknowledged here.

Reviewer #2: The manuscript shows a well organized summary of the treatment options for olfactory dysfunction after COVID-19. It would be useful for physicians in the treatment of COVID-19. The limitations of the study without meta-analysis were also shown. I wish if the minor comments would help to improve.

1. Please show the P-values in Table 6.

2. The Betamethasone Nasal Drop needs a Kaiteki position to be fully effective. Please indicate if you could find that in the reference 38.

6. PLOS authors have the option to publish the peer review history of their article (what does this mean?). If published, this will include your full peer review and any attached files.

Reviewer #1: No

Reviewer #2: No

---

## [Author Response · Author response to Decision Letter 0]

9 Feb 2023

Response to Reviewers 

Number Original comments of the reviewer Reply by the author(s) Changes done on the page number and line number

16 November 2022

1 Please ensure that your manuscript meets PLOS ONE's style requirements, including those for file naming The authors revised the manuscript as The PLOS ONE style templates. "Article title" -Line numbers 1-4.

"Affiliation" -Line numbers 7-9.

"Corresponding authorship" -Line numbers 17-18.

"Abstract" -Line numbers 34,35,40,50,62,64-66.

"Introduction" -Line number 70, 120, 121.

"Materials and methods" -Line numbers 126,128,133 (Table 1),136,137,148,157,179, 186,192, 198, 226, 230 (Table 2), 233, 248, 261.

"Results" -Line numbers 282, 284, 298, 311 (Table 3), 313, 321 (Table 4), 322 (Table 5), 324, 325, 330 (Table 6), 343, 355, 363, 371, 380, 388, 400, 401, 413, 423, 438, 447, 460, 470, 478, 492.

"Discussion" -Line number 513.

"Conclusion" -Line number 648.

"Strengths and limitations of the study" -Line number 660.

"Access to data" -Line number 688.

"Dissemination plan" -Line number 694.

"Administrative information" -Line number 704.

"Author contributions" -Line number 713.

"Acknowledgements" -Line number 724.

"Supporting information" -Line number 731.

"References" -Line numbers 739. 

2 Upon re-submitting your revised manuscript, please upload your study’s minimal underlying data set as either Supporting Information files or to a stable, public repository and include the relevant URLs, DOIs, or accession numbers within your revised cover letter. The authors uploaded our study's minimal underlying data set as a Supporting Information file. 

S4 Excel file could be shared without considering ethical and legal issues. S3 Excel file -Line number 718.

3 Your ethics statement should only appear in the Methods section of your manuscript. If your ethics statement is written in any section besides the Methods, please move it to the Methods section and delete it from any other section. Please ensure that your ethics statement is included in your manuscript, as the ethics statement entered into the online submission form will not be published alongside your manuscript. The authors have already mentioned ethical considerations in the materials and methods, as follows:

The protocol was submitted to and approved by the Ethics Committee of the School Society Community and Health (Public Health Dissertation), the UK and the Institutional Review Board with the ethical consent form. This review would not take any consensus regarding personal and sensitive issues; all selected studies were publicly accessible evidence. In the extraction and reporting of the key information, the reviewers did not maximize and minimize the effects of the treatment interventions and constructed the connected finding in an adherent manner. Line numbers 262-269. 

4 We require you to either (1) present written permission from the copyright holder to publish these figures specifically under the CC BY 4.0 license, or (2) remove the figures from your submission: The authors removed figure 2 from the manuscript. Line number 717. 

Reviewer (1)

5 Abstract: JBI? Yes, JBI (Joanna Briggs Institute) 

6 Abstract: Why is it that the authors have recommended that the potential risks of steroids should be carefully considered? What findings from their review have led to this recommendation? The authors deleted this recommendation as it was not relevant" Line numbers 64-65. 

7 Introduction-Line 76-current coronavirus: Be more specific. 

 Current coronavirus has been specified as " The current coronavirus (Covid-19)" Line number 75.

8 Methods: PICOT? Yes, PICOT and its details have already been explained in Table 1. Line number 131.

9 Line 140: Delete “Surely,”. Surely was deleted as per the reviewer's instruction. Line number 141.

10 Line 150: Is “abbreviation” a type of publication? Yes, a type of publication without full information. 

11 Line 163: Pregnancy? Pregnant? Pregnancy, it was corrected as "pregnant" Line number 173.

12 Describe/define “mild”, “moderate” and “severe” COVID-19 patients either in the ‘introduction’ section or preferably in the ‘methods’ section as it had a bearing in the inclusion/exclusion criteria formulated by the authors of the present review. The authors defined mild, moderate and severe forms of Covid-19 as (Mild Covid-19 was defined as an individual who has one or more Covid-19-related signs and symptoms such as fever, nausea, vomiting, malaise, diarrhoea, taste and smell loss and cough without breathlessness and others abnormal chest imaging. Moderate Covid-19 was defined as an individual with the manifestations of mild Covid-19 plus abnormal chest imaging, lower respiratory disease, and pulse oximeter ((SpO2) ≥94%). The definition of a severe form of Covid-19 included SpO2 <94%, a respiration rate per minute more than 30, a percent of lung infiltrates more than 50 and PaO2 (Arterial Partial Pressure of Oxygen)/ FiO2 (Fraction of Inspired Oxygen) less than 300 mmHg.). Line numbers 158-167. 

13 Line 189: When was the search performed? The literature search was done from March 2022 to April 2022. Line number 198. 

14 Line 199: PICO? Yes, the authors used "PICO" as a search strategy. 

15 Table 2 and Line 189: PubMed, PubMed Central and EMBASE are mentioned (Line 189). MEDLINE features in Table 2. MEDLINE was deleted and substituted by EMBASE Table 2. 

16 Line 259: Research? Review? Review. 

The usage of research was deleted and substituted by "review" Line number 269. 

17 Lines 263-270, Ethical consideration: Not applicable; Delete. As per the reviewer's instruction, the authors removed "ethical consideration" from Method and material section. Line numbers 273-280.

18 Lines 275-277: Why should there be a difference in the number of items identified by two different researchers/reviewers when the same search string is applied at the same database on the same day? This is not acceptable and points towards lapses in the search strategy. Screening of the identified items is done by 2 reviewers and discrepancies sorted out in consultation with a third reviewer. Identification of items does not require 2 reviewers. Isn’t it? And why is there a discrepancy in the numbers related to identification of items? Two reviewers independently searched and screened the titles and abstracts by using the same databases and predefined search terms, but the number of titles and abstracts searched and screened by two independent reviewers was different because of the reasons

(1) A reviewer used EMBASE to search on the start date while another used PubMed on the same start date. We think it is a normal process.

(2) Although the search results that appeared on each search of a database applied could be the same, each reviewer's ability and giving time on the screening might be different. We think that is normal.

(3) Two independent reviewers made a comprehensive search by doing the additional identification of citations and references which were mentioned in the text and reference lists of the published papers. In this case, the published screened by each reviewer might not be the same, if so the number of citations and references identified by each reviewer might be different. We think that is normal. 

(4) According to the guideline of PRISMA, identifying the records by two independent reviewers is a good practice. 

19 Line 280: The total of “8767” records identified is wrong. Why are the numbers from PubMed added twice? The same is the case for PubMed Central and EMBASE. This is not appropriate. The total of "8767" records is right. This total included (1568 + 1329 + 1401 + 1449 + 1120 + 1501 + 198 + 201). Reviewer 1 (1568 + 1329 + 1401 + 198) and Reviewer 2 (1449 + 1120 + 1501 + 201). 

20 The quality of the figure related to the search strategy (PRISMA flowchart) is not good and should be replaced with a better-quality figure/picture. For related details, refer to the instructions to authors on the journal website. As per the reviewer's instruction, the authors revised the PRISMA flow chart.

" S1 Fig 1. PRISMA 2020 flow diagram for new systematic reviews which included searches of databases and registers only" Line numbers 724-725.

21 PRISMA flowchart: What is the difference between “Studies included in the review n=21” and “Reports of included studies n=21”? The authors revised these usages to be uniform. S1 Fig. PRISMA Flow Chart 

22 PRISMA flowchart: Registers (n=8767)? Registers (n=0)? Records (n = 8767) 

23 PRISMA flowchart: The number of items/records identified is incorrect for reasons mentioned before (comment related to Lines 275-277). Total record = 8767

Duplicated records removed = 7056

Records screened = 8767 – 7056 = 1711

Records excluded = 1690 (Irrelevant study Design (n=642), Opinion/Guideline/Handouts/Instructions (n=138)

- Incomplete/Protocol/During Publication (n= 426)

- Irrelevant intervention (n= 148)

- Irrelevant study outcomes (n=76)

- Not human study (n=7)

- Irrelevant study population (n=184)

- Not include intervention (n=69) 

(642 + 138 + 426 + 148 + 76 + 7 + 184 + 69 = 1690)

Studies south for retrieval = 1711 – 1690 = 21 

24 PRISMA flowchart: Records sought for retrieval (n=?)? Studies south for retrieval = (n=21) 

25 PRISMA flowchart, Identification: The number of records/items identified from citation searching is not mentioned. Again, there are two different numbers mentioned for two different reviewers in the text (Line 278). The authors mentioned the number of records/items identified from citation searching as others (n = 198) and (n = 201) respectively in S1 Fig and also explained in the text. Line numbers 286-287. 

26 S2 Figure is of poor quality and should be replaced with a better-quality one. The authors removed S2 Fig from the manuscript. 

27 Lines 284-285: Cite all the 10 RCTs. As per the reviewer's instruction, the authors cited all 10 RCTs. Line number 305

28 Lines 284-285: Cite all the 11 quasi-experimental studies. As per the reviewer's instruction, the authors cited all 11 quasi-experimental studies. Line numbers 305-306.

29 Line 289: Cite all the 21 studies included in the present systematic review. As per the reviewer's instruction, the authors cited all 21 studies included in the present systematic review. Line number 293. 

30 Line 354: “In the trial of (Singh, Jain and Parveen, 2021)”? Yes, the authors corrected these words like " In the trial of Singh, Jain and Parveen, 2021". Line number 362.

31 Covid-19 vaccination: Why is this considered as a treatment strategy in the present systematic review? It shouldn’t be included. The related studies that the present authors have included have studied the occurrence of anosmia in COVID-19 vaccinated individuals infected with COVID-19. Primarily, this review aimed to summarize all possible treatment options and other interventions relieving Covid-19-related anosmia and determine the effectiveness of steroids on the treatment and the duration of the Covid-19-related olfactory dysfunction compared to that of other medications and interventions. This review also found the effects of Covid-19 vaccination on Covid-19-related anosmia as an intervention. So, let us mention the effects of Covid-19 vaccination on Covid-19-related anosmia in our manuscript. 

32 Line 503: Replace “Discussion on main findings” with “Discussion”. As per the reviewer's instruction, the authors replaced "Discussion on main findings" with "Discussion". Line number 511

33 Line 509: Cite the previous systematic reviews and meta-analyses that are here referred to by the present authors. Besides, mention how was the present systematic review different in terms of the different parameters listed by the present authors. A Table depicting the list of related previous systematic reviews (plus the present systematic review) and the differences should be included. This is important to highlight the fact that the present systematic review is not redundant and the importance of publishing the present systematic review. As per the reviewer's instruction, the authors added a supplementary table file mentioning the different parameters listed by present studies included in this review, but the authors were difficult to construct a comparative table showing the differences in findings between this review and previous reviews because the authors have not found the previous review focusing on similar settings. Please see: Supplementary table (S 4 Table Objectives and measurement tools of the included studies)

34 Lines 603-609: There is a combination of Vancouver and Harvard referencing styles. Here and elsewhere, corrections should be made sticking to the Vancouver style. As per the reviewer's instructions, the authors changed Harvard referencing styles to Vancouver. Line numbers 528, 570, 601, 605, 615, 616, 617, 629, 630, 635, 638. 

35 Lines 645-646: On what basis is it recommended that the potential risks of steroids should be carefully considered? What findings in the present systematic review have led the present authors to stating this recommendation? The authors deleted this recommendation as it was not relevant" Line number 653-654. 

36 Line 669: Delete “desperately”. As per the reviewer's instruction, the authors deleted "desperately". 

37 Lines 692-696 – Administrative information: Mention who is the student awarded the Master’s degree and the administrative roles of all others included as co-authors. Hasn’t the supervisor fulfilled the authorship criteria? The supervisor seems to be acknowledged here. As per the reviewer's instruction, the authors revised "Administrative information" like " The utilization of unauthorized intellectual properties is refrained by the authors. The whole paper is free from copying or stealing the ideas or artworks of other scholars, authors, and professional co-workers. This was the student dissertation work of Dr Phyo Zin Win which has been undertaken for the award of Master of Science in Public Health from the University of Bedfordshire (UK). The co-reviewer and supervisor had the responsibilities for designing the review, undertaking all processes of the systematic review and revising the manuscript together with the principal investigator. Another co-author had the responsibilities for selecting the studies per protocol, extracting the relevant data, and reviewing and editing the manuscript." Line number 701-707. 

Reviewer (2)

1 Please show the P-values in Table 6. As per the reviewer's instruction, the authors added all P-values found in the studies into Table 6. Line number 330. 

2 The Betamethasone Nasal Drop needs a Kaiteki position to be fully effective. Please indicate if you could find that in the reference 38. As per the reviewer's recommendation, the author found this position in the study and mentioned this position in this review. Line number 357. 

Additional Editor Comments 

1 Please delete Ref 1 and 7, which is not peer-reviewed article. As per the editor's comment. The authors changed references 1 and 7 into " Abu.Farha, M., Thanaraj TA., Qaddoumi MG., Hashem A, Abubaker J and Al-Mulla F. The role of lipid metabolism in Covid-19 virus infection and as a drug target. Int. J. Mol. Sci. 2020, 21, 3544; doi:10.3390/ijms21103544 and Lucia Iannuzzi, Anna Eugenia Salzo, Gioacchino Angarano, Vincenzo Ostilio Palmieri, Piero Portincasa, Annalisa Saracino, Matteo Gelardi, Michele Dibattista, Nicola Quaranta, Gaining Back What Is Lost: Recovering the Sense of Smell in Mild to Moderate Patients After COVID-19, Chemical Senses, Volume 45, Issue 9, November 2020, Pages 875–881, https://doi.org/10.1093/chemse/bjaa066

Line numbers 739-741 and line numbers 761-765.

2 World Health Organization (WHO) first declared the coronavirus outbreak as a global health emergency on 30 January 2020 -> Please cite the article such as DOI: https://doi.org/10.54724/lc.2022.e10

As per the editor's comment, the author added DOI: https://doi.org/10.54724/lc.2022.e10 to reference 3. 

Line number 751. 

3 Please change Ref 35 to 2020 PRISMA guideline (DOI: https://doi.org/10.54724/lc.2022.e9). As per the editor's comment, the authors changed reference 35 to "2020 PRISMA guideline (DOI: https://doi.org/10.54724/lc.2022.e9)."

Line number 860.

8 February 2023

Additional Comments

1 Please ensure that you refer/cite to Table 2 & 6 in your text as, if accepted, production will need this reference to link the reader to the Table. As per the 1st comment, the authors filled references to each table. Line number 218, and line number 310

2 We note that your author list was updated during the revision process. Please complete our Authorship Change Form by following this link: http://journals.plos.org/plosone/s/file?id=13d0/plos-one-change-to-authorship-form.docx If you are adding or removing more than 2 authors, you can complete multiple forms and submit as many forms as needed to reflect the updates. Please return the form(s) as an attachment by emailing plosone@plos.org or by uploading it as a submission file labeled with the file type ‘Other’. Please note that if your manuscript is accepted, we will not be able to complete the publication process without the completed form. As per the 2nd comment, information about the authors was updated by using Authorship Change Form of PLOS ONE. No author was removed or added but certain information about them was slightly changed for accuracy. The form was separately submitted along with the manuscript by labelling it as the file type ‘Other’.

---

## [Decision Letter · Decision Letter 1]

3 Apr 2023

PONE-D-22-27243R1Effect of any form of steroids in comparison with that of other medications on the duration of olfactory dysfunction in covid-19 patients: a systematic review of randomized trials and quasi-experimental studiesPLOS ONE

Dear Dr. Winn,

Thank you for submitting your manuscript to PLOS ONE. After careful consideration, we feel that it has merit but does not fully meet PLOS ONE’s publication criteria as it currently stands. Therefore, we invite you to submit a revised version of the manuscript that addresses the points raised during the review process.

We look forward to receiving your revised manuscript.

Kind regards,

Dong Keon Yon, MD, FACAAI

Academic Editor

PLOS ONE

Additional Editor Comments :

Still, the reviewers have raised a number of very important issues, and their excellent comments will need to be adequately addressed in a revision before the acceptability of your manuscript for publication in the Journal can be determined.

#1. I mean that please delete citation [3] and replace this peer-reviewed article DOI: https://doi.org/10.54724/lc.2022.e10

Reviewers' comments:

Reviewer's Responses to Questions

**Comments to the Author**

1. If the authors have adequately addressed your comments raised in a previous round of review and you feel that this manuscript is now acceptable for publication, you may indicate that here to bypass the “Comments to the Author” section, enter your conflict of interest statement in the “Confidential to Editor” section, and submit your "Accept" recommendation.

Reviewer #1: (No Response)

Reviewer #2: All comments have been addressed

2. Is the manuscript technically sound, and do the data support the conclusions?

Reviewer #1: No

Reviewer #2: Yes

3. Has the statistical analysis been performed appropriately and rigorously? 

Reviewer #1: N/A

Reviewer #2: N/A

4. Have the authors made all data underlying the findings in their manuscript fully available?

Reviewer #1: Yes

Reviewer #2: Yes

5. Is the manuscript presented in an intelligible fashion and written in standard English?

Reviewer #1: Yes

Reviewer #2: Yes

6. Review Comments to the Author

Reviewer #1: • Some of the reviewers’ comments are not incorporated (or not appropriately incorporated) in the revised draft (except that a reply is provided in the “authors’ response to the reviewers’ comments” form.

• “Abbreviations” cannot be a type of manuscript as responded by the authors.

• For a systematic review, search for the relevant articles cannot be mentioned as “from March 2022 to April 2022”. The specific date of search must be mentioned for systematic reviews. The data (articles/documents) in the databases would change on a daily basis.

• Overall, the methods employed for the search strategy of this particular systematic review are flawed. An appropriate search strategy forms the foundation for a systematic review.

Reviewer #2: The review article is useful for clinicians treating patients with COVID-19. It has been properly revised according to the comments. I have no further comments.

7. PLOS authors have the option to publish the peer review history of their article (what does this mean?). If published, this will include your full peer review and any attached files.

Reviewer #1: No

Reviewer #2: No

---

## [Author Response · Author response to Decision Letter 1]

3 May 2023

4 April 2023

1 #1. I mean that please delete citation [3] and replace this peer-reviewed article DOI: https://doi.org/10.54724/lc.2022.e10

As per the reviewers' comments, the authors deleted the citation [3] and replaced the recommended article DOI: https://doi.org/10.54724/lc.2022.e10

2 “Abbreviations” cannot be a type of manuscript as responded by the authors. The authors mentioned the meaning of abbreviations used in the text as follows.

-PubMed (Public/Publisher MEDLINE)

-EMBASE (Excerpta Medica Database)

-JBI (Joanna Briggs Institute)

-UK (United Kingdom)

-US (United States)

-PCR (Polymerase Chain Reaction)

-SPO2 (Saturation of peripheral Oxygen) 

-RCT (Randomized Controlled Clinical Trials)

-TDI (Transition Dyspnea Index)

-UPSIT (University of Pennsylvania Smell Identification Test)

-BD (Twice a day)

-OD (Once a day)

-NaCl (Sodium Chloride)

3 For a systematic review, search for the relevant articles cannot be mentioned as “from March 2022 to April 2022”. The specific date of search must be mentioned for systematic reviews. The data (articles/documents) in the databases would change on a daily basis. The author mentioned the duration of the search for relevant articles as "from 21 March 2022 to 30 April 2022"

4 Overall, the methods employed for the search strategy of this particular systematic review are flawed. An appropriate search strategy forms the foundation for a systematic review. To be more strategic search, the authors added the following information under the sub-heading "Information sources and search strategy"

" In searching the eligible studies in PubMed and PubMed Central, the following search terms were applied by two reviewers. 

"Treatments" OR Interventions" AND "Covid-19-related olfactory dysfunction":

"Treatment"[Mesh] OR "Treatment"[All Fields] OR "Medication"[All Fields] OR "Therapy"[All Fields] OR "Administration"[Mesh] OR "Management"[All Fields] OR "Care of"[All Fields] OR "Cure"[Mesh] OR "Prescription"[All Fields] OR "Regimen"[All Fields] OR "Oral corticosteroid"[All Fields] OR "Intranasal corticosteroid"[All Fields] OR "Systemic corticosteroid"[All Fields] OR "Intranasal sodium citrate"[All Fields] OR "Smell training"[All Fields] OR "Taste training"[All Fields] OR "Olfactory training"[All Fields] OR "Essential oil"[All Fields] OR "Diets"[All Fields] OR "Supplements"[All Fields] OR "Intranasal Vitamin A"[All Fields] OR "Systemic Omega-3"[All Fields] OR "Coffee"{All Fields] OR "Lavender Syrup"[Mesh] OR "Lavender Syrups"[All Fields] OR "Vaccination"[Mesh] OR "Inflawell Syrup"[Mesh] OR "Inflawell Syrup"[All Fields] OR "Carbamazepine"[Mesh] OR "Palmitoylethanolamide"[Mesh] OR "PEA"[Mesh] OR "Luteolin"[Mesh] AND "Covid-19-related olfactory dysfunction"[Mesh] OR "covid-19-related olfactory dysfunctions"[TIAB] OR "covid-19-associated olfactory dysfunction"[TIAB] OR "Covid-19-related smell loss"[Mesh] OR "Covid-19-related odor loss"[All Fields] OR "Covid-19-related loss of smell"[All Fields] OR "Covid-19-related loss of odor"[All Fields] OR "Covid-19-associated smell loss"[Mesh] OR "Covid-19-associated odor loss"[All Fields] OR "Covid-19-associated loss of smell"[All Fields] OR "Covid-19-associated loss of odor"[All Fields] OR "anosmia"[All Fields] OR

"parosmia"[All Fields] OR "olfactory dysfunction"[All Fields] OR "olfactory disorder"[All Fields] 

"Effects of Treatment" AND "Corticosteroid" AND "Covid-19-related olfactory dysfunction":

"Effects of Treatment"[Mesh] OR "Effectiveness of Treatment"[All Fields] OR "Treatment effectiveness"[All Fields] OR "Treatment efficacy"[All Fields] OR "Strengths of Treatment"[All Fields] AND "Corticosteroid"[Mesh] OR "Steroid"[All Fields] OR "Steroids"[All Fields] OR "Delivering Corticosteroid treatment"[Mesh] OR "Administrating Corticosteroid treatment "[All Fields] OR "Indexing Corticosteroid treatment "[All Fields] OR "Dosing Corticosteroid treatment "[All Fields] OR "Delivering Steroid treatment"[All Fields] OR "Administrating Steroid treatment "[Mesh] OR "Indexing Steroid treatment "[All Fields] OR "Dosing Steroid treatment "[Mesh] AND "Covid-19-related olfactory dysfunction"[Mesh] OR "covid-19-related olfactory dysfunctions"[TIAB] OR "covid-19-associated olfactory dysfunction"[TIAB] OR "Covid-19-related smell loss"[Mesh] OR "Covid-19-related odor loss"[All Fields] OR "Covid-19-related loss of smell"[All Fields] OR "Covid-19-related loss of odor"[All Fields] OR "Covid-19-associated smell loss"[Mesh] OR "Covid-19-associated odor loss"[All Fields] OR "Covid-19-associated loss of smell"[All Fields] OR "Covid-19-associated loss of odor"[All Fields] OR "anosmia"[Mesh] OR "parosmia"[Mesh] OR "olfactory dysfunction"[Mesh] OR "olfactory disorder"[Mesh]

In searching the includable studies in EMBASE, the Quick Search and PICO Search were exercised. 

For a quick search, the following key terms were applied.

'Treatments' OR 'Interventions' AND 'Covid-19-related olfactory dysfunction':

'Treatment'/de OR 'treatment' OR 'tx' OR 'Medication' OR 'Therapy' OR 'Administration' OR 'Management' OR 'Mx' OR 'Care of'/exp OR 'Cure' OR 'Prescription'/exp OR 'Regimen' OR 'Oral corticosteroid'/exp OR 'Intranasal corticosteroid' OR 'Systemic corticosteroid' OR 'Intranasal sodium citrate'/exp OR 'Smell training'/exp OR 'Taste training' OR 'Olfactory training' OR 'Essential oil'/exp OR 'Diets'/exp OR 'Supplements'/exp OR 'Intranasal Vitamin A'/exp OR 'Systemic Omega-3' OR 'Coffee' OR 'Lavender Syrup' OR 'Vaccination' OR 'Inflawell Syrup' OR 'Carbamazepine' OR 'Palmitoylethanolamide'/PEA OR 'Luteolin' AND 'Covid-19-related olfactory dysfunction'/de OR 'covid-19-related olfactory dysfunctions'/exp OR 'covid-19-associated olfactory dysfunction'/exp OR 'Covid-19-related smell loss'/exp OR 'Covid-19-related odor loss'/exp OR 'Covid-19-related loss of smell'/exp OR 'Covid-19-related loss of odor'/exp OR 'Covid-19-associated smell loss'/de OR 'Covid-19-associated odor loss'/exp OR 'Covid-19-associated loss of smell'/exp OR 'Covid-19-associated loss of odor'/exp OR 'anosmia'/de OR 'parosmia'/exp OR 'olfactory dysfunction'/de OR 'olfactory disorder' 

"Effects of Treatment" AND "Corticosteroid" AND "Covid-19-related olfactory dysfunction":

'Effects of Treatment'/de OR 'Effectiveness of Treatment' OR 'tx eff' OR 'Treatment effectiveness' OR 'Treatment efficacy' OR 'Strengths of Treatment' AND 'Corticosteroid'/de OR 'Steroid'/exp OR 'Steroids'/exp OR 'Delivering Corticosteroid treatment'/de OR 'Administrating Corticosteroid treatment'/exp OR 'Indexing Corticosteroid treatment'/exp OR 'Dosing Corticosteroid treatment'/exp OR 'Delivering Steroid treatment'/exp OR 'Administrating Steroid treatment'/exp OR 'Indexing Steroid treatment'/exp OR 'Dosing Steroid treatment'/exp AND 'Covid-19-related olfactory dysfunction'/de OR 'covid-19-related olfactory dysfunctions'/exp OR 'covid-19-associated olfactory dysfunction'/exp OR 'Covid-19-related smell loss'/exp OR 'Covid-19-related odor loss'/exp OR 'Covid-19-related loss of smell'/exp OR 'Covid-19-related loss of odor'/exp OR 'Covid-19-associated smell loss'/de OR 'Covid-19-associated odor loss'/exp OR 'Covid-19-associated loss of smell'/exp OR 'Covid-19-associated loss of odor'/exp OR 'anosmia'/de OR 'parosmia'/exp OR 'olfactory dysfunction'/de OR 'olfactory disorder'"

5 If published, this will include your full peer review and any attached files.

If published, the authors want to include our full peer review and S1 Fig. S2 Table, and S4 Table but do not want to include S3 Table.

---

## [Decision Letter · Decision Letter 2]

24 May 2023

PONE-D-22-27243R2Effect of any form of steroids in comparison with that of other medications on the duration of olfactory dysfunction in covid-19 patients: a systematic review of randomized trials and quasi-experimental studiesPLOS ONE

Dear Dr. Winn,

Thank you for submitting your manuscript to PLOS ONE. After careful consideration, we feel that it has merit but does not fully meet PLOS ONE’s publication criteria as it currently stands. Therefore, we invite you to submit a revised version of the manuscript that addresses the points raised during the review process.

We look forward to receiving your revised manuscript.

Kind regards,

Dong Keon Yon, MD, FACAAI

Academic Editor

PLOS ONE

Journal Requirements:

Additional Editor Comments:

This is an excellent paper. However, the authors have to receive English Editing Service.

#1. Covid-19 -> COVID-19

#2. COVID-19 pateitns -> patients with COVID-19

#3. Zinc -> zinc

#4. Palmitoylethanolamide -> palmitoylethanolamide

#5. Luteolin and Omega-3 -> luteolin and omega-3

#6. d PCR (Polymerase Chain Reaction)-confirmed SARS-CoV-2 -> polymerase chain reaction (PCR)-confirmed SARS-CoV-2

#7. ((SpO2)-Saturation of peripheral Oxygen ≥94%). -> (saturation of peripheral oxygen [SpO2] ≥94%).

#8. PaO2 (Arterial Partial Pressure of Oxygen) -> arterial partial pressure of oxygen(PaO2)

..

..

..

Reviewers' comments:

Reviewer's Responses to Questions

**Comments to the Author**

1. If the authors have adequately addressed your comments raised in a previous round of review and you feel that this manuscript is now acceptable for publication, you may indicate that here to bypass the “Comments to the Author” section, enter your conflict of interest statement in the “Confidential to Editor” section, and submit your "Accept" recommendation.

Reviewer #2: All comments have been addressed

2. Is the manuscript technically sound, and do the data support the conclusions?

Reviewer #2: Yes

3. Has the statistical analysis been performed appropriately and rigorously? 

Reviewer #2: N/A

4. Have the authors made all data underlying the findings in their manuscript fully available?

Reviewer #2: Yes

5. Is the manuscript presented in an intelligible fashion and written in standard English?

Reviewer #2: Yes

6. Review Comments to the Author

Reviewer #2: My comments have been appropriately corrected. This paper has been shown to be more accurate. I have no further comments to offer.

7. PLOS authors have the option to publish the peer review history of their article (what does this mean?). If published, this will include your full peer review and any attached files.

Reviewer #2: No

---

## [Author Response · Author response to Decision Letter 2]

21 Jun 2023

Response to Reviewers 

Number Original comments of the reviewer Reply by the author(s) Changes done on the page number and line number

16 November 2022

1 Please ensure that your manuscript meets PLOS ONE's style requirements, including those for file naming The authors revised the manuscript as The PLOS ONE style templates. "Article title" -Line numbers 1-4.

"Affiliation" -Line numbers 7-9.

"Corresponding authorship" -Line numbers 17-18.

"Abstract" -Line numbers 34,35,40,50,62,64-66.

"Introduction" -Line number 70, 120, 121.

"Materials and methods" -Line numbers 126,128,133 (Table 1),136,137,148,157,179, 186,192, 198, 226, 230 (Table 2), 233, 248, 261.

"Results" -Line numbers 282, 284, 298, 311 (Table 3), 313, 321 (Table 4), 322 (Table 5), 324, 325, 330 (Table 6), 343, 355, 363, 371, 380, 388, 400, 401, 413, 423, 438, 447, 460, 470, 478, 492.

"Discussion" -Line number 513.

"Conclusion" -Line number 648.

"Strengths and limitations of the study" -Line number 660.

"Access to data" -Line number 688.

"Dissemination plan" -Line number 694.

"Administrative information" -Line number 704.

"Author contributions" -Line number 713.

"Acknowledgements" -Line number 724.

"Supporting information" -Line number 731.

"References" -Line numbers 739. 

2 Upon re-submitting your revised manuscript, please upload your study’s minimal underlying data set as either Supporting Information files or to a stable, public repository and include the relevant URLs, DOIs, or accession numbers within your revised cover letter. The authors uploaded our study's minimal underlying data set as a Supporting Information file. 

S4 Excel file could be shared without considering ethical and legal issues. S3 Excel file -Line number 718.

3 Your ethics statement should only appear in the Methods section of your manuscript. If your ethics statement is written in any section besides the Methods, please move it to the Methods section and delete it from any other section. Please ensure that your ethics statement is included in your manuscript, as the ethics statement entered into the online submission form will not be published alongside your manuscript. The authors have already mentioned ethical considerations in the materials and methods, as follows:

The protocol was submitted to and approved by the Ethics Committee of the School Society Community and Health (Public Health Dissertation), the UK and the Institutional Review Board with the ethical consent form. This review would not take any consensus regarding personal and sensitive issues; all selected studies were publicly accessible evidence. In the extraction and reporting of the key information, the reviewers did not maximize and minimize the effects of the treatment interventions and constructed the connected finding in an adherent manner. Line numbers 262-269. 

4 We require you to either (1) present written permission from the copyright holder to publish these figures specifically under the CC BY 4.0 license, or (2) remove the figures from your submission: The authors removed figure 2 from the manuscript. Line number 717. 

Reviewer (1)

5 Abstract: JBI? Yes, JBI (Joanna Briggs Institute) 

6 Abstract: Why is it that the authors have recommended that the potential risks of steroids should be carefully considered? What findings from their review have led to this recommendation? The authors deleted this recommendation as it was not relevant" Line numbers 64-65. 

7 Introduction-Line 76-current coronavirus: Be more specific. 

 Current coronavirus has been specified as " The current coronavirus (Covid-19)" Line number 75.

8 Methods: PICOT? Yes, PICOT and its details have already been explained in Table 1. Line number 131.

9 Line 140: Delete “Surely,”. Surely was deleted as per the reviewer's instruction. Line number 141.

10 Line 150: Is “abbreviation” a type of publication? Yes, a type of publication without full information. 

11 Line 163: Pregnancy? Pregnant? Pregnancy, it was corrected as "pregnant" Line number 173.

12 Describe/define “mild”, “moderate” and “severe” COVID-19 patients either in the ‘introduction’ section or preferably in the ‘methods’ section as it had a bearing in the inclusion/exclusion criteria formulated by the authors of the present review. The authors defined mild, moderate and severe forms of Covid-19 as (Mild Covid-19 was defined as an individual who has one or more Covid-19-related signs and symptoms such as fever, nausea, vomiting, malaise, diarrhoea, taste and smell loss and cough without breathlessness and others abnormal chest imaging. Moderate Covid-19 was defined as an individual with the manifestations of mild Covid-19 plus abnormal chest imaging, lower respiratory disease, and pulse oximeter ((SpO2) ≥94%). The definition of a severe form of Covid-19 included SpO2 <94%, a respiration rate per minute more than 30, a percent of lung infiltrates more than 50 and PaO2 (Arterial Partial Pressure of Oxygen)/ FiO2 (Fraction of Inspired Oxygen) less than 300 mmHg.). Line numbers 158-167. 

13 Line 189: When was the search performed? The literature search was done from March 2022 to April 2022. Line number 198. 

14 Line 199: PICO? Yes, the authors used "PICO" as a search strategy. 

15 Table 2 and Line 189: PubMed, PubMed Central and EMBASE are mentioned (Line 189). MEDLINE features in Table 2. MEDLINE was deleted and substituted by EMBASE Table 2. 

16 Line 259: Research? Review? Review. 

The usage of research was deleted and substituted by "review" Line number 269. 

17 Lines 263-270, Ethical consideration: Not applicable; Delete. As per the reviewer's instruction, the authors removed "ethical consideration" from Method and material section. Line numbers 273-280.

18 Lines 275-277: Why should there be a difference in the number of items identified by two different researchers/reviewers when the same search string is applied at the same database on the same day? This is not acceptable and points towards lapses in the search strategy. Screening of the identified items is done by 2 reviewers and discrepancies sorted out in consultation with a third reviewer. Identification of items does not require 2 reviewers. Isn’t it? And why is there a discrepancy in the numbers related to identification of items? Two reviewers independently searched and screened the titles and abstracts by using the same databases and predefined search terms, but the number of titles and abstracts searched and screened by two independent reviewers was different because of the reasons

(1) A reviewer used EMBASE to search on the start date while another used PubMed on the same start date. We think it is a normal process.

(2) Although the search results that appeared on each search of a database applied could be the same, each reviewer's ability and giving time on the screening might be different. We think that is normal.

(3) Two independent reviewers made a comprehensive search by doing the additional identification of citations and references which were mentioned in the text and reference lists of the published papers. In this case, the published screened by each reviewer might not be the same, if so the number of citations and references identified by each reviewer might be different. We think that is normal. 

(4) According to the guideline of PRISMA, identifying the records by two independent reviewers is a good practice. 

19 Line 280: The total of “8767” records identified is wrong. Why are the numbers from PubMed added twice? The same is the case for PubMed Central and EMBASE. This is not appropriate. The total of "8767" records is right. This total included (1568 + 1329 + 1401 + 1449 + 1120 + 1501 + 198 + 201). Reviewer 1 (1568 + 1329 + 1401 + 198) and Reviewer 2 (1449 + 1120 + 1501 + 201). 

20 The quality of the figure related to the search strategy (PRISMA flowchart) is not good and should be replaced with a better-quality figure/picture. For related details, refer to the instructions to authors on the journal website. As per the reviewer's instruction, the authors revised the PRISMA flow chart.

" S1 Fig 1. PRISMA 2020 flow diagram for new systematic reviews which included searches of databases and registers only" Line numbers 724-725.

21 PRISMA flowchart: What is the difference between “Studies included in the review n=21” and “Reports of included studies n=21”? The authors revised these usages to be uniform. S1 Fig. PRISMA Flow Chart 

22 PRISMA flowchart: Registers (n=8767)? Registers (n=0)? Records (n = 8767) 

23 PRISMA flowchart: The number of items/records identified is incorrect for reasons mentioned before (comment related to Lines 275-277). Total record = 8767

Duplicated records removed = 7056

Records screened = 8767 – 7056 = 1711

Records excluded = 1690 (Irrelevant study Design (n=642), Opinion/Guideline/Handouts/Instructions (n=138)

- Incomplete/Protocol/During Publication (n= 426)

- Irrelevant intervention (n= 148)

- Irrelevant study outcomes (n=76)

- Not human study (n=7)

- Irrelevant study population (n=184)

- Not include intervention (n=69) 

(642 + 138 + 426 + 148 + 76 + 7 + 184 + 69 = 1690)

Studies south for retrieval = 1711 – 1690 = 21 

24 PRISMA flowchart: Records sought for retrieval (n=?)? Studies south for retrieval = (n=21) 

25 PRISMA flowchart, Identification: The number of records/items identified from citation searching is not mentioned. Again, there are two different numbers mentioned for two different reviewers in the text (Line 278). The authors mentioned the number of records/items identified from citation searching as others (n = 198) and (n = 201) respectively in S1 Fig and also explained in the text. Line numbers 286-287. 

26 S2 Figure is of poor quality and should be replaced with a better-quality one. The authors removed S2 Fig from the manuscript. 

27 Lines 284-285: Cite all the 10 RCTs. As per the reviewer's instruction, the authors cited all 10 RCTs. Line number 305

28 Lines 284-285: Cite all the 11 quasi-experimental studies. As per the reviewer's instruction, the authors cited all 11 quasi-experimental studies. Line numbers 305-306.

29 Line 289: Cite all the 21 studies included in the present systematic review. As per the reviewer's instruction, the authors cited all 21 studies included in the present systematic review. Line number 293. 

30 Line 354: “In the trial of (Singh, Jain and Parveen, 2021)”? Yes, the authors corrected these words like " In the trial of Singh, Jain and Parveen, 2021". Line number 362.

31 Covid-19 vaccination: Why is this considered as a treatment strategy in the present systematic review? It shouldn’t be included. The related studies that the present authors have included have studied the occurrence of anosmia in COVID-19 vaccinated individuals infected with COVID-19. Primarily, this review aimed to summarize all possible treatment options and other interventions relieving Covid-19-related anosmia and determine the effectiveness of steroids on the treatment and the duration of the Covid-19-related olfactory dysfunction compared to that of other medications and interventions. This review also found the effects of Covid-19 vaccination on Covid-19-related anosmia as an intervention. So, let us mention the effects of Covid-19 vaccination on Covid-19-related anosmia in our manuscript. 

32 Line 503: Replace “Discussion on main findings” with “Discussion”. As per the reviewer's instruction, the authors replaced "Discussion on main findings" with "Discussion". Line number 511

33 Line 509: Cite the previous systematic reviews and meta-analyses that are here referred to by the present authors. Besides, mention how was the present systematic review different in terms of the different parameters listed by the present authors. A Table depicting the list of related previous systematic reviews (plus the present systematic review) and the differences should be included. This is important to highlight the fact that the present systematic review is not redundant and the importance of publishing the present systematic review. As per the reviewer's instruction, the authors added a supplementary table file mentioning the different parameters listed by present studies included in this review, but the authors were difficult to construct a comparative table showing the differences in findings between this review and previous reviews because the authors have not found the previous review focusing on similar settings. Please see: Supplementary table (S 4 Table Objectives and measurement tools of the included studies)

34 Lines 603-609: There is a combination of Vancouver and Harvard referencing styles. Here and elsewhere, corrections should be made sticking to the Vancouver style. As per the reviewer's instructions, the authors changed Harvard referencing styles to Vancouver. Line numbers 528, 570, 601, 605, 615, 616, 617, 629, 630, 635, 638. 

35 Lines 645-646: On what basis is it recommended that the potential risks of steroids should be carefully considered? What findings in the present systematic review have led the present authors to stating this recommendation? The authors deleted this recommendation as it was not relevant" Line number 653-654. 

36 Line 669: Delete “desperately”. As per the reviewer's instruction, the authors deleted "desperately". 

37 Lines 692-696 – Administrative information: Mention who is the student awarded the Master’s degree and the administrative roles of all others included as co-authors. Hasn’t the supervisor fulfilled the authorship criteria? The supervisor seems to be acknowledged here. As per the reviewer's instruction, the authors revised "Administrative information" like " The utilization of unauthorized intellectual properties is refrained by the authors. The whole paper is free from copying or stealing the ideas or artworks of other scholars, authors, and professional co-workers. This was the student dissertation work of Dr Phyo Zin Win which has been undertaken for the award of Master of Science in Public Health from the University of Bedfordshire (UK). The co-reviewer and supervisor had the responsibilities for designing the review, undertaking all processes of the systematic review and revising the manuscript together with the principal investigator. Another co-author had the responsibilities for selecting the studies per protocol, extracting the relevant data, and reviewing and editing the manuscript." Line number 701-707. 

Reviewer (2)

1 Please show the P-values in Table 6. As per the reviewer's instruction, the authors added all P-values found in the studies into Table 6. Line number 330. 

2 The Betamethasone Nasal Drop needs a Kaiteki position to be fully effective. Please indicate if you could find that in the reference 38. As per the reviewer's recommendation, the author found this position in the study and mentioned this position in this review. Line number 357. 

Additional Editor Comments 

1 Please delete Ref 1 and 7, which is not peer-reviewed article. As per the editor's comment. The authors changed references 1 and 7 into " Abu.Farha, M., Thanaraj TA., Qaddoumi MG., Hashem A, Abubaker J and Al-Mulla F. The role of lipid metabolism in Covid-19 virus infection and as a drug target. Int. J. Mol. Sci. 2020, 21, 3544; doi:10.3390/ijms21103544 and Lucia Iannuzzi, Anna Eugenia Salzo, Gioacchino Angarano, Vincenzo Ostilio Palmieri, Piero Portincasa, Annalisa Saracino, Matteo Gelardi, Michele Dibattista, Nicola Quaranta, Gaining Back What Is Lost: Recovering the Sense of Smell in Mild to Moderate Patients After COVID-19, Chemical Senses, Volume 45, Issue 9, November 2020, Pages 875–881, https://doi.org/10.1093/chemse/bjaa066

Line numbers 739-741 and line numbers 761-765.

2 World Health Organization (WHO) first declared the coronavirus outbreak as a global health emergency on 30 January 2020 -> Please cite the article such as DOI: https://doi.org/10.54724/lc.2022.e10

As per the editor's comment, the author added DOI: https://doi.org/10.54724/lc.2022.e10 to reference 3. 

Line number 751. 

3 Please change Ref 35 to 2020 PRISMA guideline (DOI: https://doi.org/10.54724/lc.2022.e9).

As per the editor's comment, the authors changed reference 35 to "2020 PRISMA guideline (DOI: https://doi.org/10.54724/lc.2022.e9)."

Line number 860. 

8 February 2023

Additional Comments

1 Please ensure that you refer/cite to Table 2 & 6 in your text as, if accepted, production will need this reference to link the reader to the Table. As per the 1st comment, the authors filled references to each table. Line number 218, and line number 310

2 We note that your author list was updated during the revision process. Please complete our Authorship Change Form by following this link: http://journals.plos.org/plosone/s/file?id=13d0/plos-one-change-to-authorship-form.docx If you are adding or removing more than 2 authors, you can complete multiple forms and submit as many forms as needed to reflect the updates. Please return the form(s) as an attachment by emailing plosone@plos.org or by uploading it as a submission file labeled with the file type ‘Other’. Please note that if your manuscript is accepted, we will not be able to complete the publication process without the completed form. As per the 2nd comment, information about the authors was updated by using Authorship Change Form of PLOS ONE. No author was removed or added but certain information about them was slightly changed for accuracy. The form was separately submitted along with the manuscript by labelling it as the file type ‘Other’. 

4 April 2023

1 #1. I mean that please delete citation [3] and replace this peer-reviewed article DOI: https://doi.org/10.54724/lc.2022.e10

As per the reviewers' comments, the authors deleted the citation [3] and replaced the recommended article DOI: https://doi.org/10.54724/lc.2022.e10

Line numbers 798-805. 

2 “Abbreviations” cannot be a type of manuscript as responded by the authors. The authors mentioned the meaning of abbreviations used in the text as follows.

-PubMed (Public/Publisher MEDLINE)

-EMBASE (Excerpta Medica Database)

-JBI (Joanna Briggs Institute)

-UK (United Kingdom)

-US (United States)

-PCR (Polymerase Chain Reaction)

-SPO2 (Saturation of peripheral Oxygen) 

-RCT (Randomized Controlled Clinical Trials)

-TDI (Transition Dyspnea Index)

-UPSIT (University of Pennsylvania Smell Identification Test)

-BD (Twice a day)

-OD (Once a day)

-NaCl (Sodium Chloride) 

Line number 34

Line number 34

Line number 38

Line number 88

Line number 89

Line number 152

Line number 158

Line number 326

Line number 367 (Table 3)

Line number 367 (Table 3)

Line number 386 (Table 6)

Line number 386 (Table 6)

Line number 415

3 For a systematic review, search for the relevant articles cannot be mentioned as “from March 2022 to April 2022”. The specific date of search must be mentioned for systematic reviews. The data (articles/documents) in the databases would change on a daily basis. The author mentioned the duration of the search for relevant articles as "from 21 March 2022 to 30 April 2022" Line number 191

4 Overall, the methods employed for the search strategy of this particular systematic review are flawed. An appropriate search strategy forms the foundation for a systematic review. To be more strategic search, the authors added the following information under the sub-heading "Information sources and search strategy"

" In searching the eligible studies in PubMed and PubMed Central, the following search terms were applied by two reviewers. 

"Treatments" OR Interventions" AND "Covid-19-related olfactory dysfunction":

"Treatment"[Mesh] OR "Treatment"[All Fields] OR "Medication"[All Fields] OR "Therapy"[All Fields] OR "Administration"[Mesh] OR "Management"[All Fields] OR "Care of"[All Fields] OR "Cure"[Mesh] OR "Prescription"[All Fields] OR "Regimen"[All Fields] OR "Oral corticosteroid"[All Fields] OR "Intranasal corticosteroid"[All Fields] OR "Systemic corticosteroid"[All Fields] OR "Intranasal sodium citrate"[All Fields] OR "Smell training"[All Fields] OR "Taste training"[All Fields] OR "Olfactory training"[All Fields] OR "Essential oil"[All Fields] OR "Diets"[All Fields] OR "Supplements"[All Fields] OR "Intranasal Vitamin A"[All Fields] OR "Systemic Omega-3"[All Fields] OR "Coffee"{All Fields] OR "Lavender Syrup"[Mesh] OR "Lavender Syrups"[All Fields] OR "Vaccination"[Mesh] OR "Inflawell Syrup"[Mesh] OR "Inflawell Syrup"[All Fields] OR "Carbamazepine"[Mesh] OR "Palmitoylethanolamide"[Mesh] OR "PEA"[Mesh] OR "Luteolin"[Mesh] AND "Covid-19-related olfactory dysfunction"[Mesh] OR "covid-19-related olfactory dysfunctions"[TIAB] OR "covid-19-associated olfactory dysfunction"[TIAB] OR "Covid-19-related smell loss"[Mesh] OR "Covid-19-related odor loss"[All Fields] OR "Covid-19-related loss of smell"[All Fields] OR "Covid-19-related loss of odor"[All Fields] OR "Covid-19-associated smell loss"[Mesh] OR "Covid-19-associated odor loss"[All Fields] OR "Covid-19-associated loss of smell"[All Fields] OR "Covid-19-associated loss of odor"[All Fields] OR "anosmia"[All Fields] OR

"parosmia"[All Fields] OR "olfactory dysfunction"[All Fields] OR "olfactory disorder"[All Fields] 

"Effects of Treatment" AND "Corticosteroid" AND "Covid-19-related olfactory dysfunction":

"Effects of Treatment"[Mesh] OR "Effectiveness of Treatment"[All Fields] OR "Treatment effectiveness"[All Fields] OR "Treatment efficacy"[All Fields] OR "Strengths of Treatment"[All Fields] AND "Corticosteroid"[Mesh] OR "Steroid"[All Fields] OR "Steroids"[All Fields] OR "Delivering Corticosteroid treatment"[Mesh] OR "Administrating Corticosteroid treatment "[All Fields] OR "Indexing Corticosteroid treatment "[All Fields] OR "Dosing Corticosteroid treatment "[All Fields] OR "Delivering Steroid treatment"[All Fields] OR "Administrating Steroid treatment "[Mesh] OR "Indexing Steroid treatment "[All Fields] OR "Dosing Steroid treatment "[Mesh] AND "Covid-19-related olfactory dysfunction"[Mesh] OR "covid-19-related olfactory dysfunctions"[TIAB] OR "covid-19-associated olfactory dysfunction"[TIAB] OR "Covid-19-related smell loss"[Mesh] OR "Covid-19-related odor loss"[All Fields] OR "Covid-19-related loss of smell"[All Fields] OR "Covid-19-related loss of odor"[All Fields] OR "Covid-19-associated smell loss"[Mesh] OR "Covid-19-associated odor loss"[All Fields] OR "Covid-19-associated loss of smell"[All Fields] OR "Covid-19-associated loss of odor"[All Fields] OR "anosmia"[Mesh] OR "parosmia"[Mesh] OR "olfactory dysfunction"[Mesh] OR "olfactory disorder"[Mesh]

In searching the includable studies in EMBASE, the Quick Search and PICO Search were exercised. 

For a quick search, the following key terms were applied.

'Treatments' OR 'Interventions' AND 'Covid-19-related olfactory dysfunction':

'Treatment'/de OR 'treatment' OR 'tx' OR 'Medication' OR 'Therapy' OR 'Administration' OR 'Management' OR 'Mx' OR 'Care of'/exp OR 'Cure' OR 'Prescription'/exp OR 'Regimen' OR 'Oral corticosteroid'/exp OR 'Intranasal corticosteroid' OR 'Systemic corticosteroid' OR 'Intranasal sodium citrate'/exp OR 'Smell training'/exp OR 'Taste training' OR 'Olfactory training' OR 'Essential oil'/exp OR 'Diets'/exp OR 'Supplements'/exp OR 'Intranasal Vitamin A'/exp OR 'Systemic Omega-3' OR 'Coffee' OR 'Lavender Syrup' OR 'Vaccination' OR 'Inflawell Syrup' OR 'Carbamazepine' OR 'Palmitoylethanolamide'/PEA OR 'Luteolin' AND 'Covid-19-related olfactory dysfunction'/de OR 'covid-19-related olfactory dysfunctions'/exp OR 'covid-19-associated olfactory dysfunction'/exp OR 'Covid-19-related smell loss'/exp OR 'Covid-19-related odor loss'/exp OR 'Covid-19-related loss of smell'/exp OR 'Covid-19-related loss of odor'/exp OR 'Covid-19-associated smell loss'/de OR 'Covid-19-associated odor loss'/exp OR 'Covid-19-associated loss of smell'/exp OR 'Covid-19-associated loss of odor'/exp OR 'anosmia'/de OR 'parosmia'/exp OR 'olfactory dysfunction'/de OR 'olfactory disorder' 

"Effects of Treatment" AND "Corticosteroid" AND "Covid-19-related olfactory dysfunction":

'Effects of Treatment'/de OR 'Effectiveness of Treatment' OR 'tx eff' OR 'Treatment effectiveness' OR 'Treatment efficacy' OR 'Strengths of Treatment' AND 'Corticosteroid'/de OR 'Steroid'/exp OR 'Steroids'/exp OR 'Delivering Corticosteroid treatment'/de OR 'Administrating Corticosteroid treatment'/exp OR 'Indexing Corticosteroid treatment'/exp OR 'Dosing Corticosteroid treatment'/exp OR 'Delivering Steroid treatment'/exp OR 'Administrating Steroid treatment'/exp OR 'Indexing Steroid treatment'/exp OR 'Dosing Steroid treatment'/exp AND 'Covid-19-related olfactory dysfunction'/de OR 'covid-19-related olfactory dysfunctions'/exp OR 'covid-19-associated olfactory dysfunction'/exp OR 'Covid-19-related smell loss'/exp OR 'Covid-19-related odor loss'/exp OR 'Covid-19-related loss of smell'/exp OR 'Covid-19-related loss of odor'/exp OR 'Covid-19-associated smell loss'/de OR 'Covid-19-associated odor loss'/exp OR 'Covid-19-associated loss of smell'/exp OR 'Covid-19-associated loss of odor'/exp OR 'anosmia'/de OR 'parosmia'/exp OR 'olfactory dysfunction'/de OR 'olfactory disorder'"

 Line numbers 197-268

5 If published, this will include your full peer review and any attached files.

If published, the authors want to include our full peer review and S1 Fig. S2 Table, and S4 Table but do not want to include S3 Table. 

26 May 2023

1 Please review your reference list to ensure that it is complete and correct. If you have cited papers that have been retracted, please include the rationale for doing so in the manuscript text, or remove these references and replace them with relevant current references. Any changes to the reference list should be mentioned in the rebuttal letter that accompanies your revised manuscript. If you need to cite a retracted article, indicate the article’s retracted status in the References list and also include a citation and full reference for the retraction notice. As recommended by the editor, the authors checked and rechecked the citations in the text and the reference lists. After checking, 

Reference number 3 (Kim SY, Yeniova AZR. Global, regional, and national incidence and mortality of COVID-19 in 237 countries and territories, January 2022: a systematic analysis for World Health Organization COVID-19 Dashboard. Life Cycle 2022;2:e10

https://doi.org/10.54724/lc.2022.e10) was replaced by Kim SY, Yeniova AÖ. Global, regional, and national incidence and mortality of COVID-19; 237 countries and territories, January 2022: A systematic analysis for world health organization COVID-19 dashboard. Life Cycle. 2022;2. doi:10.54724/lc.2022.e10.

Reference number 10 (Myshchenko I, Ostrovskyy M, Kolhanov A, Makoida I and Hrechukh L. PREVALENCE OF SENSORY DYSFUNCTIONS IN ADULT UKRAINIAN POPULATION WITH LABORATORY CONFIRMED COVID-19. Wiad Lek. 2022; 75(3): 670-677. PMID: 35522877.) was replaced by Myshchenko I, Ostrovskyy M, Kolhanov A, Makoida I, Hrechukh L. Prevalence of sensory dysfunctions in adult Ukrainian population with laboratory confirmed COVID-19. Wiadomości Lekarskie. 2022;75(3):670–677. doi:10.36740/wlek202203119.

Reference number 35 (2020 PRISMA guideline (DOI: https://doi.org/10.54724/lc.2022.e9) was replaced by Page MJ, McKenzie JE, Bossuyt PM, Boutron I, Hoffmann TC, Mulrow CD, et al. The Prisma 2020 statement: An updated guideline for reporting systematic reviews. BMJ. 2021; doi:10.1136/bmj.n71.

Reference number 56 (Altundag A, Cayonu M, Kayabasoglu G, Salihoglu M, Tekeli H, Saglam O, et al. Modified olfactory training in patients with postinfectious olfactory loss. Laryngoscope. 2015: 125(8): 1763-6. doi: 10.1002/lary.25245. Epub 2015 Jun 2. PMID: 26031472.) was replaced by Altundag A, Yilmaz E, Kesimli MC. Modified olfactory training is an effective treatment method for Covid‐19 induced parosmia. The Laryngoscope. 2022;132(7):1433–8. doi:10.1002/lary.30101.

Reference number 57 (Harless L and Liang J. Pharmacologic treatment for post-viral olfactory dysfunction: a systematic review. International Forum of Allergy and Rhinology 2016: 6(7): 760-767.) was replaced by Harless L, Liang J. Pharmacologic treatment for postviral olfactory dysfunction: A systematic review. International Forum of Allergy & Rhinology. 2016;6(7):760–767. doi:10.1002/alr.21727.

Reference number 59 (Huang T, Wei Y and Wu D. Effects of olfactory training on post-traumatic olfactory dysfunction: A systematic review and meta-analysis. International Forum of Allergy and Rhinology 2021: 11(7):1102-1112.) was replaced by Huang T, Wei Y, Wu D. Effects of olfactory training on posttraumatic olfactory dysfunction: A systematic review and meta‐analysis. International Forum of Allergy & Rhinology. 2021;11(7):1102–1112. doi:10.1002/alr.22758. 

Line numbers 820-826.

Line numbers 846-851.

Line numbers 933-935.

Line numbers 1017-1021.

Line numbers 1022-1025.

Line numbers 1029-1033.

2 #1. Covid-19 -> COVID-19

#2. COVID-19 patients -> patients with COVID-19

#3. Zinc -> zinc

#4. Palmitoylethanolamide -> palmitoylethanolamide

#5. Luteolin and Omega-3 -> luteolin and omega-3

#6. d PCR (Polymerase Chain Reaction)-confirmed SARS-CoV-2 -> polymerase chain reaction (PCR)-confirmed SARS-CoV-2

#7. ((SpO2)-Saturation of peripheral Oxygen ≥94%). -> (saturation of peripheral oxygen [SpO2] ≥94%).

#8. PaO2 (Arterial Partial Pressure of Oxygen) -> arterial partial pressure of oxygen (PaO2) As recommended by the editor, the authors changed some words in the manuscript. Whole manuscript

---

## [Editor Report · Decision Letter 3]

26 Jun 2023

Effect of any form of steroids in comparison with that of other medications on the duration of olfactory dysfunction in patients with COVID-19: a systematic review of randomized trials and quasi-experimental studies

PONE-D-22-27243R3

Dear Dr. Winn,

We’re pleased to inform you that your manuscript has been judged scientifically suitable for publication and will be formally accepted for publication once it meets all outstanding technical requirements.

Kind regards,

Dong Keon Yon, MD, FACAAI, FAAAAI

Academic Editor

PLOS ONE

Additional Editor Comments (optional):

This is an excellent paper.
---

## [Editor Report · Acceptance letter]

6 Jul 2023

PONE-D-22-27243R3 

Effect of any form of steroids in comparison with that of other medications on the duration of olfactory dysfunction in patients with COVID-19: a systematic review of randomized trials and quasi-experimental studies 

Dear Dr. Winn:

I'm pleased to inform you that your manuscript has been deemed suitable for publication in PLOS ONE. Congratulations! Your manuscript is now with our production department. 

Kind regards, 

on behalf of

Dr. Dong Keon Yon 

Academic Editor

PLOS ONE